# Fantastic Features and Where to Find Them: A Probing Method to combine Features from Multiple Foundation Models

**Benjamin Ramtoula**[1]    **Pierre-Yves Lajoie**[2]    **Paul Newman**[1]    **Daniele De Martini**[1]

[1]University of Oxford    [2]Polytechnique Montréal

{benjamin, pnewman, daniele}@robots.ox.ac.uk

pierre-yves.lajoie@polymtl.ca

## Abstract

Foundation models (FMs) trained with different objectives and data learn diverse representations, making some more effective than others for specific downstream tasks. Existing adaptation strategies, such as parameter-efficient fine-tuning, focus on individual models and do not exploit the complementary strengths across models. Probing methods offer a promising alternative by extracting information from frozen models, but current techniques do not scale well with large feature sets and often rely on dataset-specific hyperparameter tuning. We propose Combined backBones (ComBo), a simple and scalable probing-based adapter that effectively integrates features from multiple models and layers. ComBo compresses activations from layers of one or more FMs into compact token-wise representations and processes them with a lightweight transformer for task-specific prediction. Crucially, ComBo does not require dataset-specific tuning or backpropagation through the backbone models. However, not all models are equally relevant for all tasks. To address this, we introduce a mechanism that leverages ComBo's joint multi-backbone probing to efficiently evaluate each backbone's task-relevance, enabling both practical model comparison and improved performance through selective adaptation. On the 19 tasks of the VTAB-1k benchmark, ComBo outperforms previous probing methods, matches or surpasses more expensive alternatives, such as distillation-based model merging, and enables efficient probing of tuned models. Our results demonstrate that ComBo offers a practical and general-purpose framework for combining diverse representations from multiple FMs.

## 1   Introduction

Transfer learning of foundation models (FMs) has become the predominant approach to solve new downstream tasks in computer vision. Through large-scale pre-training, these models develop general representations, useful across diverse tasks. Practitioners now have access to multiple such models, each trained with specific data and supervision: CLIP [1] to align text and image embeddings, MAE [2] to reconstruct missing image patches, or SAM [3] to segment regions in images. These differences in training objectives and data distribution influence the learned representations, making specific models more suitable than others for downstream tasks (Fig. 1a).

Ideally, the complementary representations of these models should be harnessed when solving a new task, but it is unclear how to combine these different models. Whilst effective, tuning techniques are usually designed to be applied to individual models and often require backpropagating through the backbone. Trial-and-error of tuning techniques to find the best model would mean that users need to

---

Project page: bramtoula.github.io/combo

39th Conference on Neural Information Processing Systems (NeurIPS 2025).

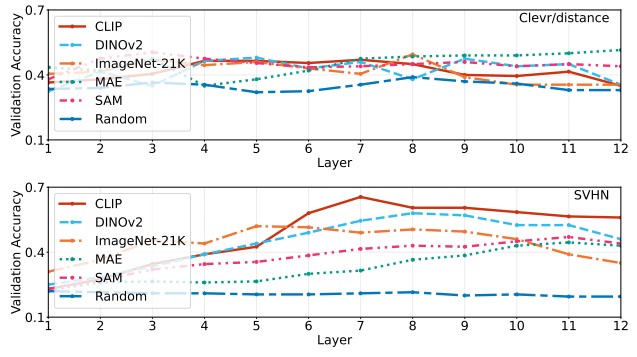

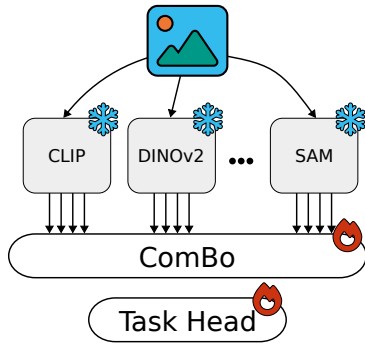

(a) Validation accuracy of linear probing on activations from different layers of vision transformers (ViTs) trained from CLIP [1], DINOv2 [6], ImageNet-21K [7], MAE [2], SAM [3], as well as a randomly initialised ViT. The experiments are on the Clevr/distance [8] and SVHN [9] tasks of the VTAB-1k benchmark [10].

(b) Combined backBones (ComBo) adapter overview. Our adapter uses activations from multiple layers of multiple models to solve downstream tasks.

Figure 1: **(left)** Different foundation models learn diverse representations, and given different downstream tasks, the most capable pre-trained model might not be the same. Not only can the model vary, but the location of the layer containing the most directly relevant features can also change, with intermediate layers occasionally leading to the highest probing accuracy. **(right)** We propose an adapter that can take advantage of these diverse representations by probing layer outputs from multiple frozen models. This allows us to efficiently adapt to diverse downstream tasks without needing to backpropagate through potentially large foundation models.

backpropagate through every model to select the best one, which can quickly become computationally expensive as models become larger and more models of interest are released. Other approaches studied in recent works [4, 5] distil multiple FMs into a single student model. However, this is an expensive process during which it can be challenging to preserve all useful features from all teachers.

Probing-based adapters [11, 12] leverage activations from multiple layers of frozen pre-trained models, making them well-suited for combining representations from multiple models without backpropagation. Unfortunately, probing-based adapters still underperform compared to tuning methods and require careful regularisation. Moreover, current probing approaches do not scale well to activations from dense feature maps and typically resort to average-pooling neighbouring activations to reduce the number of inputs, with pooling region sizes specifically tuned for each dataset.

We propose Combined backBones (ComBo) adapter, a novel probing-based adapter that not only overcomes the limitations of prior methods but also enables the combination of activations from multiple models (Fig. 1b). We design our adapter to process raw feature maps from pre-trained models without arbitrary pooling, relying only on learnt parameters to compress information. ComBo works by combining embeddings from different layers into a compact descriptor for each original image patch and training a small transformer to process these descriptors. ComBo scales to multiple input models and consistently outperforms previous probing techniques, without requiring extensive task-specific hyperparameter or regularisation tuning, all while maintaining strong performance and efficiency when applied to individual models. While different models can complement each other, specific subsets of models will likely be more relevant for each task. As such, we also introduce a method using ComBo to efficiently measure each model's task-relevance, providing a practical way for practitioners to select the most relevant backbones for a task. We find that using the most relevant combination of backbones can also improve ComBo's efficiency and performance.

Our **contributions** are as follows: 1) An analysis of differences between representations learned by foundation models (FMs), 2) A new probing adapter for pre-trained FMs that demonstrates greater stability and performance than previous probing approaches, 3) Practical demonstration of multi-model integration capabilities: probing multiple large FMs together without backpropagation through any of them, making large model combinations accessible to practitioners with limited computational resources, and 4) A method using ComBo to compare the relevance of backbones for given downstream tasks, enabling efficient model selection for ComBo and other adaptation methods.

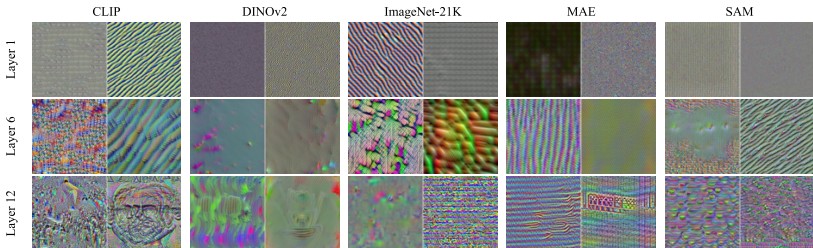

Figure 2: Images that maximise activations of different neurons of different models [1–3, 6, 7], using the technique from Ghiasi et al. [13]. All models rely on a ViT-B architecture, but each was trained with different data and supervision. Although all models usually pick up low-level patterns in early layers, distinct patterns appear as we get to intermediate and later layers, highlighting differences in learned representations. Models trained with ImageNet-21K, MAE, or SAM appear more sensitive to specific textures in later layers, whereas late CLIP neurons appear to react to patterns of semantic entities, and DINOv2 late neurons appear more sensitive to complex abstract shapes. These observations align with the idea that diverse features can be found by using multiple models, but also by using multiple layer outputs. Images are generated from randomly sampled neurons.

## 2 Related works

**Representations learned by FMs**    FMs are models trained on large amounts of diverse data and that can effectively be adapted to diverse downstream tasks [14]. Examples include DINOv2 [6, 15], CLIP [1], and SAM [3], all based on the ViT architecture [16]. Although there are indications that large-scale models trained with different modalities and supervisions are converging towards a "platonic representation" [17], current generations of models still display variations in their representations, and consequently on their suitability for given downstream tasks. In fact, multiple previous studies have observed variations in learned representations based on the type of supervision of a model [13, 18–20]. We further validate these studies by comparing the performance of modern FMs by linear probing outputs of different layers of different models, shown in Fig. 1a, and by applying the visualisation technique from Ghiasi et al. [13] to produce results shown in Fig. 2.

**Combining multiple FMs**    Given the complementary strengths of different FMs, several works combine features from multiple pre-trained models. In meta-learning, Chowdhury et al. [21] show that using a "library" of multiple pre-trained diverse feature extractors significantly outperforms single extractors. Liu et al. [22] propose a "Universal Representation Transformer" that uses attention to dynamically combine features from multiple models, achieving strong few-shot learning performance. However, both only use final layer outputs, ignoring intermediate layers that provide complementary information but require adapter architectures capable of handling the increased feature dimensionality. Other works focus on model merging [23], combining task-specific models without additional training by merging weights [24–26]. However, the constraint of not using any additional data is not always present, and improvements can be gained by efficiently using training to combine models. This setting is explored with agglomerative models such as SAM-CLIP [4] and RADIO [27], which use knowledge distillation [28] to train a student model retaining multiple FMs strengths. However, distillation is expensive and must be repeated for each additional FM, and can struggle to preserve all models fairly, with issues such as "mode switch" where the distilled model aligns more with specific teachers at certain resolutions [5].

**Tuning vision models**    Regardless of whether multiple FMs are combined or used separately, adapting pre-trained models to new downstream tasks requires careful consideration of adaptation techniques, which we categorise into two groups: tuning-based and probing-based methods. Tuning-based approaches involve updating model weights to modify their behaviour in the forward pass. While full fine-tuning can be expensive, Parameter-efficient fine-tuning (PEFT) techniques, such as adapters [29], LoRA [30], and prompt tuning [31], provide techniques to update models with few parameters and often outperform full fine-tuning. In particular, Adapter+ [32] has found that adapter modules [29] in specific configurations and appropriate initialisations can outperform other more complex adaptation techniques without the need for extensive task-specific tuning. However, these tuning methods are typically designed to adapt a single model rather than leverage complementary representations from multiple models simultaneously, and they usually rely on backpropagating through the adapted model, which can become prohibitive with multiple large models.

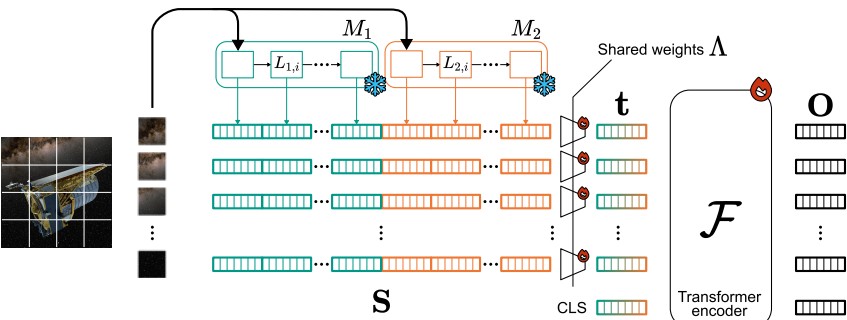

Figure 3: The ComBo adapter. Given intermediate feature maps from multiple models, we first learn a small projection $\Lambda$ of their combined layers' embeddings which we apply to all their tokens. These tokens are then passed to a small transformer $\mathcal{F}$ model which outputs a `cls` token on which we place our classification head.

**Probing vision models** Alternatively, probing-based approaches rely on extracted activations from frozen models. This approach is particularly appealing when combining multiple FMs, as it eliminates the need to backpropagate through each model. Head2Toe [11] observed the potential of probing features from multiple layers of a model with a linear layer. Building on this, Structured Model Probing (SMP) [12] noted that different tasks might require more complex adapters than a single linear layer. They improved over Head2Toe by introducing a new regularisation loss based on structures within the model; they also incorporated an MLP into the adapter architecture, which would naturally be regularised to have minimal effect on simpler tasks. While these probing approaches could theoretically be extended to incorporate input features from multiple models, they face practical limitations. They require careful regularisation and hyperparameter tuning dependent on the specific task, and they do not scale efficiently with an increasing number of input features. Both methods typically resort to average-pooling neighbouring activations to reduce computational demands, with pooling region sizes manually tuned for different datasets. In contrast, our approach extends the probing paradigm to efficiently leverage features from multiple FMs without the scaling limitations of previous methods, while eliminating the need for dataset-specific hyperparameter tuning.

## 3 Method

Prior approaches like Head2Toe and SMP [11, 12] have demonstrated the value of probing activations from multiple layers of pre-trained models. They do so by stacking activations from different layers and feature map locations and passing them through a linear layer. However, naive stacking can lead to very high-dimensional inputs: for a ViT-B with 12 blocks, each processing 197 768-dimensional tokens; stacking them and applying them as input of a 100-class linear classifier would require more than 181M parameters. Head2Toe and SMP manage this by reducing feature map resolution through task-dependent average pooling, a process that we argue can discard important spatial information.

To address this issue, we introduce ComBo, a novel probing adapter to efficiently leverage rich representations from multiple FMs while preserving spatial information. We build ComBo around a transformer architecture [33] for its proven ability to dynamically attend to relevant information through its self-attention mechanism, making it ideal for focusing on task-relevant features from and across pre-trained models. Rather than processing raw pixel values, ComBo's transformer processes features extracted from pre-trained model activations. This design allows our method to identify and focus on the most task-relevant features across different models and spatial locations, while benefiting from the extensive research and optimisations developed for transformer architectures in recent years.

Directly feeding concatenated features from multiple models and layers into a transformer would lead to prohibitive input size, especially as we include more models. To address this challenge, we use a two-stage approach that separates layer-wise feature selection from spatial feature integration. A first stage introduces an affine projection layer before the transformer to compress high-dimensional concatenated features from multiple layers at spatial locations into a lower-dimensional representation; this approach preserves spatial information by maintaining the token structure while reducing the embedding dimension. The compressed tokens are then processed by ComBo's transformer.

We now formalise our approach in the context of probing features from transformer backbones, as these are the dominant architectures in recent vision FMs. Fig. 3 depicts our system.

**Feature map extractions**  Let $\mathcal{M} = \{M_1, M_2, \ldots, M_K\}$ be a set of $K$ pre-trained models, each with $N_k$ layers $\mathcal{L}_k = \{L_{k,1}, L_{k,2}, \ldots, L_{k,N_k}\}$. Given an input image $\mathbf{x}$, each model operates according to its architecture, initially partitioning the image into patches and subsequently processing it as a sequence of tokens. At the output of each layer $l$ in model $M_k$, we probe the feature maps $\mathbf{F}_{k,l} \in \mathbb{R}^{T_k \times D_k}$, where $T_k$ is the number of processed tokens and $D_k$ is the embedding dimension of model $M_k$. Since different models might produce different numbers of tokens, we interpolate the feature maps to a consistent spatial resolution: we reshape each feature map to its 2D spatial layout and apply bilinear interpolation to obtain a uniform token count $T$ across all models. We finally normalise each interpolated feature map to ensure consistent scale across models and layers through mean and standard deviation computed across both token and embedding dimensions. We denote the final processed feature map at layer $l$ of model $k$ with the symbol $\hat{\mathbf{F}}_{k,l}$.

**Layer embeddings compression**  Across the layers, each token index $i \in \{1, 2, \ldots, T\}$ corresponds to a spatial position in the input image. We can then stack the normalised features from all selected layers of all models at the specific spatial positions by:

$$\mathbf{S}_i = [\hat{\mathbf{F}}_{1,1}[i,:]; \hat{\mathbf{F}}_{1,2}[i,:]; \ldots; \hat{\mathbf{F}}_{K,N_K}[i,:]] \in \mathbb{R}^D \tag{1}$$

where $D = \sum_{k=1}^{K} \sum_{l=1}^{N_k} D_k$ is the total dimension of the stacked features.

To learn a selection of the most relevant features among the layers and models, we learn an affine projection $\Lambda = \{\mathbf{W}, \mathbf{b}\}$ to compress the stacked features to a dimension $D' \ll D$. This projection is shared across all token positions, allowing a first selection of the most relevant features from all layers and models considered while preserving the spatial layout of feature maps.

**Transformer processing**  The compressed token representations $\mathbf{t} = \{\mathbf{t}_1, \mathbf{t}_2, \ldots, \mathbf{t}_T\}$ are then processed by a transformer encoder $\mathcal{F}$ after prepending a learnable class token $\mathbf{t}_{\mathrm{cls}}$:

$$\mathbf{O} = \mathcal{F}([\mathbf{t}_{\mathrm{cls}}; \mathbf{t}_1; \mathbf{t}_2; \ldots; \mathbf{t}_T]) \tag{2}$$

For classification tasks, we use the output class token $\mathbf{o}_{\mathrm{cls}}$ and apply a linear classifier to predict the class probabilities. When training the ComBo adapter, we jointly optimise the affine projection parameters $(\mathbf{W}, \mathbf{b})$, the learnable class token $\mathbf{t}_{\mathrm{cls}}$, and the transformer model $\mathcal{F}$.

Our approach is simple but powerful. Separating the feature selection through a first affine projection over embeddings of different layers, and then a transformer to combine spatially distributed features enables our approach to scale well to more inputs without discarding any original feature map activations. Moreover, our transformer leverages strong features from pre-trained models, enabling efficient low-dimensional embeddings while still capturing essential complex patterns.

**Evaluating task-relevance of models**  While combining multiple FMs can improve performance, not all models are equally relevant for every task. Including only the most task-relevant backbones can reduce computational costs while maintaining or improving performance. To identify relevant backbones, we introduce a mechanism inspired by the feature selection in Head2Toe [11], but adapted to select entire backbones rather than individual features. ComBo's joint processing of multiple backbones enables us to do so without evaluating each model individually.

To evaluate models' task-relevance, we train ComBo with an additional regularisation term applied to the projection weights $\mathbf{W}$ in $\Lambda$. For each backbone $M_k$, we compute an importance score $s_k$ (measuring task-relevance) as the $\ell_2$ norm of the projection weights associated with all its layers (i.e., the columns of $\mathbf{W}$ corresponding to that model's representations). We add these scores to our training loss via a regularisation term that encourages sparse model selection by penalizing the sum of scores: $\mathcal{L}_{\mathrm{total}} = \mathcal{L}_{\mathrm{task}} + \lambda \sum_{k=1}^{K} s_k$, where $\mathcal{L}_{\mathrm{task}}$ is the task-specific loss (e.g., cross-entropy) and $\lambda$ is a regularisation coefficient. After training with regularisation, the final importance scores $\{s_1, s_2, \ldots, s_K\}$ directly reflect each model's task-relevance. When selecting model subsets for ComBo, we first train with all models and regularisation to compute importance scores, then select the most relevant models and retrain without regularisation using only this subset.

**Applying ComBo to other architectures**  While our paper focuses on ViT-based architectures, which currently represent the dominant paradigm in modern vision FMs, the core ComBo methodology could likely be extended to other backbone architectures. The general principle remains: (1) extract features from multiple layers of multiple models, (2) align them spatially in an architecture-dependent manner, (3) compress them via a learned projection, and (4) process the compressed

features with a transformer. For architectures with different tokenisation strategies or spatial layouts, such as CNNs, feature maps could be handled through different spatial alignment and interpolation strategies, similar to how we manage different patch numbers in ViTs.

# 4 Experiments

**Dataset**   One of the key motivations of our approach is to have access to general features that might be useful for diverse tasks. To validate this, we perform experiments on the Visual Task Adaptation Benchmark (VTAB) [10]. VTAB consists of 19 tasks covering widely different image domains, making it ideal for validating the performance of our approach. The tasks are separated into three categories: *Natural* tasks contain classical computer vision problems (e.g., semantic classification), *Specialised* tasks focus on images from specialised equipment (e.g., satellite or medical imagery), and *Structured* tasks require understanding of the structure of the scene (e.g., object counting). All tasks are framed as classification problems, allowing us to maintain a similar architecture and metrics for assessing performance in these diverse domains. To validate the performance in low-data settings, we focus on the VTAB-1k setting with 1000 labelled images per task to train our adapter. We use the provided splits of 800 training and 200 validation images, and the full test sets for final results.

**Training**   We apply ComBo to ViT-B [16] pre-trained frozen backbones and consider feature maps at the output of each block (12 feature maps per model). For each VTAB-1k task, we process the feature maps through the ComBo adapter (Sec. 3). The `cls` token $o_{cls}$ at the output of our transformer model $\mathcal{F}$ is then fed to a linear classifier head that we train jointly with parameters of our affine projection and our transformer ($\mathcal{F}$) using a cross-entropy loss.

Where Head2Toe and SMP [11, 12] require hyperparameter tuning of feature pooling region size, optimiser settings, and regularisation strength, we find that ComBo performs well on all tasks with constant settings. As such, we follow the settings used in Adapter+ [32]: for all datasets, we use a batch size of 64 and an AdamW optimiser [34] with a learning rate of 0.001 and a weight decay of 0.0001. We train our model for 100 epochs with a linear warmup over 10 epochs and a cosine scheduler. When evaluating the task-relevance of models, we use a regularisation coefficient $\lambda$ of 0.01. We report the mean accuracy value over 3 seeds. Following the original VTAB settings and Adapter+ [10, 32], we resize images to $224 \times 224$ pixels and do not use any augmentations. For our transformer $\mathcal{F}$, we use a standard implementation from the `timm` library [35] with a depth of 6 blocks, an embedding dimension of 128, and 2 heads (1.7M parameters). When combining multiple models with different tokenisation settings (e.g., DINOv2 ViT-B/14 [6] and SigLIP ViT-B/16 [36]), we interpolate feature maps to the smallest shape. For ComBo adaptations, we discard any backbone `cls` and register tokens and only use tokens corresponding to initial image patches.

## 4.1   Probing a single backbone

In Tab. 1, we first validate the performance of ComBo when probing a single model, specifically ViT-B/16 trained on ImageNet-21K [7, 37], commonly used to evaluate adaptation methods.

Parameter-efficient tuning methods consistently outperform probing methods in our evaluation, with Adapter+ achieving the best overall performance despite using fixed hyperparameters across tasks. This is unsurprising as these methods preserve the pre-trained model's structure while efficiently updating the forward pass, making them effective even with limited training data. Probing methods are typically less performant since they face the challenge of learning optimal feature combinations from scratch, particularly in low-data regimes. Despite this difficulty, our proposed ComBo adapter outperforms both full fine-tuning and previous probing approaches.

We observe that while ComBo, Head2Toe, and SMP perform comparably on natural and specialised tasks, ComBo demonstrates substantial improvements on structured tasks. This performance gap likely stems from fundamental differences in how spatial information is processed. Previous approaches employ average pooling across feature maps, which reduces spatial resolution and potentially discards critical information for structured tasks. In contrast, ComBo initially preserves the complete original feature maps, learning which information to preserve.

A key practical advantage of ComBo is its effectiveness with fixed hyperparameters across all datasets. Unlike Head2Toe [11] and SMP [12], which require dataset-specific hyperparameter tuning, our

Table 1: **Comparing tuning and probing techniques for a ViT-B/16 trained on ImageNet-21K [7, 37] on the VTAB-1k benchmark.** We report accuracy (%) on the *test* set. ⚡ indicates methods requiring task-specific hyperparameter optimisation. Our proposed ComBo adapter outperforms other probing methods and full fine-tuning while maintaining fixed hyperparameters across all datasets, demonstrating particular strength on structured tasks where spatial information preservation is critical. PEFT methods (e.g., Adapter+) achieve the highest performance overall. We highlight the best result for each adapter category in **bold**.

| | Natural | | | | | | | | Specialised | | | | | Structured | | | | | | | | | Global Average |
| --- | --- | --- | --- | --- | --- | --- | --- | --- | --- | --- | --- | --- | --- | --- | --- | --- | --- | --- | --- | --- | --- | --- | --- |
| | Caltech101 [38] | CIFAR-100 [39] | DTD [40] | Flowers102 [41] | Pets [42] | Sun397 [43] | SVHN [44] | *Average* | Camelyon [45] | EuroSAT [46] | Resisc45 [47] | Retinopathy [48] | *Average* | Clevr-Count [8] | Clevr-Dist [8] | DMLab [49] | dSpr-Loc [50] | dSpr-Ori [50] | KITTI-Dist [51] | sNORB-Azim [52] | sNORB-Elev [52] | *Average* | |
| *Tuning methods, require backpropagating through the adapted model* | | | | | | | | | | | | | | | | | | | | | | | |
| Full | 92.6 | 73.2 | 70.4 | 97.9 | 86.2 | 39.6 | **90.6** | 78.6 | 87.1 | 96.6 | **87.5** | 74.0 | **86.3** | 66.6 | 61.0 | 49.8 | 82.6 | 51.9 | 79.7 | 33.5 | 37.0 | 57.8 | 74.2 |
| LoRA [30] | 91.7 | 83.0 | **71.6** | 99.2 | 90.9 | **56.7** | 83.8 | 82.4 | 86.2 | 95.7 | 83.5 | 71.9 | 84.3 | 77.7 | **62.3** | 49.0 | 82.2 | 51.7 | 80.2 | 31.0 | 47.0 | 60.1 | 75.6 |
| VPT-Deep ⚡[31] | 93.0 | 83.0 | 71.2 | 99.0 | **91.3** | 56.0 | 84.1 | 82.5 | 84.9 | 96.6 | 82.5 | **74.5** | 84.6 | 77.5 | 58.7 | 49.7 | 86.2 | **56.1** | 79.6 | **37.9** | **50.7** | 62.1 | 76.4 |
| Adapter+ [32] | **94.2** | **83.7** | 71.5 | **99.3** | 90.6 | 55.8 | 88.2 | **83.3** | **87.5** | **97.0** | 87.4 | 72.9 | 86.2 | **82.9** | 60.9 | **53.7** | **88.4** | 55.2 | **80.8** | 37.3 | 46.9 | **63.3** | **77.6** |
| *Probing methods, do not require backpropagating through the adapted model, extendable to probe multiple models* | | | | | | | | | | | | | | | | | | | | | | | |
| Linear probing | 88.1 | 78.1 | 69.0 | 99.1 | 90.0 | **56.9** | 36.0 | 73.9 | 79.8 | 90.7 | 73.7 | 73.7 | 79.5 | 32.4 | 30.5 | 35.9 | 11.2 | 26.2 | 61.9 | 14.3 | 24.5 | 29.6 | 61.0 |
| Head2Toe ⚡[11] | 90.9 | 75.3 | **75.0** | **99.5** | 86.1 | 50.9 | **83.5** | 80.2 | 84.4 | 95.7 | 84.4 | 74.3 | 84.7 | 51.7 | 59.4 | 44.0 | 47.1 | 40.3 | 65.6 | 32.5 | 41.1 | 47.7 | 70.9 |
| SMP ⚡[12] | 90.9 | **79.3** | 74.9 | 99.3 | **90.4** | 55.3 | 75.0 | **80.7** | 84.8 | 96.3 | 83.1 | **75.0** | **84.8** | 77.5 | 58.0 | 40.8 | 72.5 | 44.5 | 67.5 | **33.0** | **49.0** | 55.4 | 73.6 |
| ComBo (ours) | **91.3** | 76.7 | 69.8 | 99.3 | 90.0 | 49.3 | 81.4 | 79.7 | **84.9** | **96.6** | **84.6** | 71.8 | 84.5 | **81.7** | **60.0** | **46.5** | **89.0** | **47.6** | **77.9** | 28.4 | 45.1 | **59.5** | **74.6** |

Table 2: **Probing different foundation models with ComBo on VTAB-1k.** We report accuracy (%) on the *test* set. We compare probing individual foundation models with ComBo, probing all models simultaneously, probing the top two models selected based on our task-relevance estimation method, and fine-tuning a distilled model (RADIOv2.5) with Adapter+. Our multi-model ComBo approach achieves the highest global average, demonstrating that direct probing of multiple FMs can outperform approaches requiring expensive distillation, even with strong parameter efficient tuning. The best result for each task is highlighted in **bold** and the second-best is underlined.

| | Natural | | | | | | | | Specialised | | | | | Structured | | | | | | | | | Global Average |
| --- | --- | --- | --- | --- | --- | --- | --- | --- | --- | --- | --- | --- | --- | --- | --- | --- | --- | --- | --- | --- | --- | --- | --- |
| | Caltech101 | CIFAR-100 | DTD | Flowers102 | Pets | Sun397 | SVHN | *Average* | Camelyon | EuroSAT | Resisc45 | Retinopathy | *Average* | Clevr-Count | Clevr-Dist | DMLab | dSpr-Loc | dSpr-Ori | KITTI-Dist | sNORB-Azim | sNORB-Elev | *Average* | |
| *Probing individual foundation models with ComBo (ours)* | | | | | | | | | | | | | | | | | | | | | | | |
| DFN CLIP [53] | 94.0 | 70.8 | 77.2 | 97.5 | 87.9 | 44.4 | 83.1 | 79.3 | 84.3 | 96.7 | 89.1 | 73.3 | 85.8 | 87.2 | 60.7 | 51.6 | 81.9 | 50.9 | 79.6 | 30.0 | 51.1 | 61.6 | 75.6 |
| DINOv2 [6] | 94.7 | 79.4 | 78.4 | **99.7** | 91.1 | 49.3 | 85.9 | 82.6 | **86.0** | 96.2 | 88.3 | 73.2 | 85.9 | **92.6** | 60.4 | 51.8 | 88.2 | 53.3 | 82.0 | **36.4** | **55.6** | 65.0 | 77.9 |
| SAM [3] | 76.7 | 25.6 | 52.4 | 80.0 | 55.6 | 18.3 | 77.6 | 55.2 | 78.8 | 94.9 | 75.0 | 68.8 | 79.4 | 85.6 | 58.5 | 43.3 | 84.2 | 42.8 | 78.7 | 23.3 | 39.0 | 56.9 | 63.8 |
| SigLIP [36] | 94.0 | 66.6 | **79.2** | 98.7 | 92.4 | 53.1 | 79.2 | 80.5 | 83.0 | 96.0 | 89.6 | 73.2 | 85.4 | 88.8 | 60.0 | 54.5 | 85.0 | 48.9 | 81.6 | 32.8 | 48.9 | 62.6 | 76.2 |
| *Probing multiple foundation models with ComBo* | | | | | | | | | | | | | | | | | | | | | | | |
| All four models (ours) | **95.4** | 79.2 | 79.1 | 99.6 | 92.6 | 50.4 | 86.8 | 83.3 | 84.9 | 97.0 | **90.5** | **73.5** | 86.5 | 91.5 | **61.4** | 54.1 | 89.7 | 51.5 | **82.3** | 34.1 | 54.0 | 64.8 | 78.2 |
| Top two models (ours) | 95.1 | 80.5 | 78.9 | **99.7** | **93.3** | 53.4 | 87.3 | **84.0** | 85.3 | 96.9 | 89.9 | 73.3 | 86.3 | 92.0 | 60.9 | **55.0** | **90.6** | 52.7 | 82.2 | 33.9 | 55.3 | **65.3** | **78.6** |
| *Distilling all four foundation models above and tuning the resulting model with Adapter+ [32]* | | | | | | | | | | | | | | | | | | | | | | | |
| RADIOv2.5 [5] | 91.2 | **81.8** | 77.5 | 97.2 | 89.3 | **55.7** | **93.9** | 83.8 | **86.5** | **97.1** | 89.3 | **73.5** | **86.6** | 86.1 | 58.6 | 54.9 | 85.6 | **54.7** | 81.8 | 33.1 | 41.9 | 62.1 | 77.5 |

approach maintains consistent performance without task-specific optimisation. This characteristic makes ComBo more readily applicable to new tasks compared to previous probing approaches.

## 4.2 Probing multiple backbones

We now evaluate ComBo's capacity to probe multiple FMs simultaneously. As a baseline, we consider an alternative approach to combining representations from multiple FMs: distillation. Specifically, we compare against the ViT-B/16 model from the RADIOv2.5 family [5], trained through distillation to merge representations from four FMs: DFN CLIP [53], DINOv2 [6], SAM [3], and SigLIP [36]. We compare four approaches: (1) applying ComBo to each individual FM that contributed to RADIOv2.5, (2) using ComBo to probe all models simultaneously, (3) using ComBo to probe the top two models selected for each task with our task-relevance estimation mechanism (Sec. 3), and (4) fine-tuning the distilled RADIOv2.5 model with Adapter+. Results are presented in Tab. 2.

A key finding is that ComBo applied to all models simultaneously consistently matches or outperforms probing the best individual model across all tasks. However, individual model contributions vary considerably, with SAM consistently underperforming other models on VTAB-1k tasks. Our task-relevance estimation mechanism (Sec. 3) addresses this by identifying the most relevant models

Table 3: **Performance when combining ViT-B models individually tuned with Adapter+ [32] on VTAB-1k.** We report accuracy (%) on the *test* set. We compare individual adapted models against two approaches for combining them: Multi-Adapter+ (our baseline that trains adapters on all models simultaneously with a shared classification head) and ComBo (our method that probes all adapted models). While DINOv2 achieves the highest global average (79.9%), both combination approaches excel on more individual tasks (Multi-Adapter+: 8/19, ComBo: 5/19). ComBo offers comparable performance to Multi-Adapter+ with significantly lower computational requirements. Best results are in **bold** and second-best are underlined.

| | Natural | | | | | | | | Specialised | | | | | Structured | | | | | | | | | |
| --- | --- | --- | --- | --- | --- | --- | --- | --- | --- | --- | --- | --- | --- | --- | --- | --- | --- | --- | --- | --- | --- | --- | --- |
| | Caltech101 | CIFAR-100 | DTD | Flowers102 | Pets | Sun397 | SVHN | *Average* | Camelyon | EuroSAT | Resisc45 | Retinopathy | *Average* | Clevr-Count | Clevr-Dist | DMLab | dSpr-Loc | dSpr-Ori | KITTI-Dist | sNORB-Azim | sNORB-Elev | *Average* | Global Average |
| *Individual adapted models* | | | | | | | | | | | | | | | | | | | | | | | |
| DFN CLIP [53] | 94.5 | 76.2 | 78.9 | 98.1 | 89.9 | 54.8 | 93.1 | 83.7 | 87.3 | 97.1 | 90.3 | 73.8 | 87.1 | 92.0 | 60.7 | **59.9** | 88.2 | 56.0 | 82.7 | 41.2 | 53.2 | 66.7 | 79.2 |
| DINOv2 [6] | 92.9 | 83.5 | 78.7 | **99.7** | 93.3 | 55.7 | 92.8 | 85.2 | **89.2** | 96.9 | 88.4 | 73.9 | 87.1 | 91.1 | 61.5 | 58.5 | 86.7 | 57.8 | 83.0 | **43.9** | 56.8 | 67.4 | **79.9** |
| SAM [3] | 75.2 | 32.6 | 57.7 | 74.7 | 51.4 | 18.6 | 87.1 | 56.7 | 82.9 | 95.4 | 72.7 | 72.1 | 80.8 | 81.2 | 60.1 | 45.0 | 78.1 | 46.7 | 81.3 | 28.1 | 47.0 | 58.4 | 65.3 |
| SigLIP [36] | 92.0 | 70.0 | 78.8 | 97.9 | 91.3 | 51.6 | 92.7 | 82.0 | 85.5 | 97.3 | 89.3 | 75.0 | 86.8 | **92.8** | 61.8 | 59.1 | **90.6** | 55.7 | 83.0 | 42.1 | **60.4** | **68.2** | 79.0 |
| *Approaches combining models above* | | | | | | | | | | | | | | | | | | | | | | | |
| Multi-Adapter+ | 95.1 | **83.8** | **80.9** | **99.7** | **93.8** | **57.8** | 93.1 | **86.3** | 87.5 | 97.2 | **91.7** | **76.2** | **88.2** | 79.9 | 54.9 | 56.6 | 80.0 | **59.3** | 82.6 | 32.2 | 47.7 | 61.6 | 78.7 |
| ComBo (ours) | **95.2** | 78.9 | 78.7 | 99.4 | 90.8 | 46.3 | **93.9** | 83.3 | 88.4 | **97.4** | 89.4 | 74.4 | 87.4 | 90.3 | **62.6** | 58.2 | 87.4 | 56.2 | **85.3** | 41.2 | 57.5 | 67.3 | 79.3 |

per task. Selecting the top two backbones by importance score for each task achieves the strongest overall performance (78.6% global average), improving upon the all-models approach (78.2%) while requiring fewer backbones to process. Thus, while probing all models is preferable to any single model, selecting only the most relevant backbones further improves performance.

Comparing against distillation, both ComBo variants (all four models: 78.2%; top two: 78.6%) outperform RADIOv2.5 with Adapter+ (77.5%). While results are comparable on natural and specialised tasks, ComBo demonstrates a consistent advantage on structured tasks (64.8% and 65.3% vs. 62.1%). ComBo achieves these competitive results without requiring the computationally expensive distillation process necessary for RADIOv2.5. This performance gap may stem from limitations in the distillation process, either incomplete capture of teacher representations or capacity constraints imposed by the combination in a single ViT-B/16 architecture. A key advantage of our approach is that ComBo ensures direct access to the original representations from each FM, preserving their complementary features. Furthermore, ComBo offers greater flexibility than distillation-based approaches, allowing for seamless incorporation of new models.

## 4.3 Probing tuned backbones

As established in Sec. 4.1, parameter-efficient tuning methods like Adapter+ produce powerful task-specific representations. Here, we explore how ComBo performs when applied to FMs that have already been tuned for specific tasks, addressing two questions: (1) whether ComBo can effectively combine multiple tuned models, and (2) which layers are most informative in tuned models.

**Combining multiple tuned backbones**    Building on these findings, we investigate how effectively ComBo can combine multiple FMs already individually tuned to specific tasks. We compare our approach against a strong baseline, Multi-Adapter+, which jointly fine-tunes all models with Adapter+ and combines their outputs through a trained linear head. While both approaches use tuned models, Multi-Adapter+ requires joint optimisation with backpropagation through all models during training, making it computationally expensive but potentially powerful.

Results in Tab. 3 compare tuned individual models with Multi-Adapter+ and our ComBo approach. DI-NOv2 with Adapter+ achieves the highest average accuracy (79.9%), showing strong general-purpose performance. However, it leads in only 3 of 19 tasks, suggesting its strength lies in consistency, unlike combination methods that can be influenced by weaker models on certain tasks.

Examining the combination approaches, Multi-Adapter+ excels on 8 of 19 tasks and achieves the highest performance on natural tasks (86.3%), while ComBo wins on 5 of 19 tasks and demonstrates superior performance on structured tasks (67.3% vs. 61.6%). Overall, ComBo achieves a global average accuracy (79.3%) that outperforms Multi-Adapter+ (78.7%) and approaches DINOv2 (79.9%), confirming its effectiveness despite using a more efficient combination strategy.

Table 4: **Effect of probing all layers versus only the last layer with ComBo, using the original ViT-B/16 trained on ImageNet-21K [7, 37] and versions adapted with Adapter+ [32]**. We report VTAB-1k accuracies (%) on the *val* set. Results demonstrate that while probing all layers is critical for non-adapted models, adapted models concentrate task-relevant information in the final layer.

|  | Probing the original model | | | | Probing the tuned model | | | |
| --- | --- | --- | --- | --- | --- | --- | --- | --- |
|  | Natural | Specialised | Structured | **Average** | Natural | Specialised | Structured | **Average** |
| Last layer | 72.2 | 77.6 | 41.1 | 63.6 | 82.8 | 87.1 | 60.7 | 76.9 |
| All layers | 82.1 | 87.0 | 60.5 | 76.5 | 82.1 | 87.0 | 60.5 | 76.5 |

These results demonstrate that ComBo, despite working with independently tuned models rather than jointly optimised ones, offers comparable performance to Multi-Adapter+. This is significant given that ComBo requires only forward passes of pre-tuned models, making it more efficient and scalable when dealing with large FMs or when combining models. The performance gap between combination methods and individual models like DINOv2 also indicates potential for further improvements in how ComBo manages models of varying quality, such as with our task-relevance evaluation mechanism.

**Importance of probing different layers in tuned backbones** We also investigate the relative importance of different layers when probing tuned models. In Tab. 4, we compare the performance of ComBo when applied to either all layers or only the last layer of a ViT-B/16 model, with and without prior adaptation using Adapter+. While probing all layers is crucial for non-tuned models, this advantage disappears for models tuned with Adapter+. This finding aligns with the intuition that tuning processes concentrate task-relevant features in the final layers where classification heads are applied. This entails that the efficiency of ComBo can be improved when probing multiple tuned models, as restricting probing to models' final layer reduces computational overhead without sacrificing performance.

## 4.4 Selecting subsets of backbones

In Tab. 5, we validate our model importance scoring method from Sec. 3 by probing models with ComBo, where models are selected with different strategies from the four FMs from Sec. 4.2. We establish an upper bound with an exhaustive search over all model combinations. As naive selection strategies, we report performance using all models, only DINOv2 (the best single model), and random pairs (averaged over all possible pairs). We compare these against selections based on our method's produced importance scores: top-1, top-2, and top-3 models. The large gap between exhaustive search and baselines highlights the importance of good model selection. Our importance scores consistently outperform naive strategies, with top-2 performing best overall when using a fixed number of models $n$ across tasks. However, the ideal number of models varies by task. By selecting the optimal $n$ individually for each task from our importance rankings (best per-task top-$n$), performance approaches the exhaustive search upper bound, validating the ability of our method to estimate models' task-relevance. Details of the models' importance scores and per-task optimal $n$ are available in the appendix (Sec. B.7).

Table 5: **Validation of models' task-relevance estimation**. Accuracy (%) on VTAB-1k *val* set using different strategies to select from 4 ViT-B backbones (DFN CLIP, DINOv2, SAM, SigLIP) to probe with ComBo. Best per-task top-$n$ consists in selecting $n$ leading to the best performance for each task.

|  | Nat. | Spec. | Struct. | Avg. |
| --- | --- | --- | --- | --- |
| *Using exhaustive search* | | | | |
| Best combinations (upper bound) | 83.6 | 88.1 | 64.0 | 78.6 |
| *Using naive selection* | | | | |
| All 4 models | 81.8 | 86.4 | 60.3 | 76.2 |
| Only DINOv2 (best single) | 80.7 | 86.2 | 61.1 | 76.0 |
| Random 2 models (avg.) | 80.2 | 86.2 | 59.7 | 75.4 |
| *Using our estimated importance score* | | | | |
| Top-1 | 81.0 | 86.5 | 61.2 | 76.2 |
| Top-2 | 82.8 | 87.0 | 60.1 | 76.7 |
| Top-3 | 82.6 | 86.1 | 61.1 | 76.6 |
| Best per-task top-$n$ | 83.3 | 87.4 | 63.7 | 78.1 |

## 4.5 Design validation

Tab. 6 evaluates key components of the ComBo adapter. We validate that feature map normalisation is essential, as its removal causes performance decreases in all task categories. Architecture comparisons validate the value of using a transformer, as replacing it with a linear layer or the MLP architecture used in SMP [12] leads to important drops in performance.

Table 6: **Validation of ComBo design choices**. We report accuracy (%) on the VTAB-1k *val* set when adapting the ViT-B/16 model trained on ImageNet-21K.

|  | Nat. | Spec. | Struct. | Avg. |
| --- | --- | --- | --- | --- |
| ComBo adapter | 79.3 | 85.6 | 56.5 | 73.8 |
| w/o feature map normalisation | 79.2 | 84.9 | 50.5 | 71.5 |
| *Replacing the transformer* | | | | |
| w/ a linear head | 67.4 | 82.6 | 45.4 | 65.1 |
| w/ the MLP from SMP [12] | 71.6 | 83.2 | 47.8 | 67.5 |

Table 7: **Comparison of adapter sizes, computational costs, and memory requirements** when adapting ViT-B models. We report number of parameters and FLOPs relative to full fine-tuning, and peak GPU memory usage. Results are reported for processing a batch of 32 images of size $224 \times 224$, which was the highest power of 2 that fit in our GPU memory for all methods.

| | Size | Train | | Inference | |
|---|---|---|---|---|---|
| Method | Updated params | Rel. FLOPs | Peak GPU mem. | Rel. FLOPs | Peak GPU mem. |
| *Adapting one backbone* | | | | | |
| Fine-Tuning | 100.00% | 100.00% | 4.736 GB | 100.00% | 0.595 GB |
| Linear Probing | 0.01% | 33.18% | 0.603 GB | 100.00% | 0.595 GB |
| Adapter+ | 0.38% | 65.58% | 2.993 GB | 100.33% | 0.596 GB |
| ComBo adapter | 2.79% | 35.78% | 0.929 GB | 103.00% | 0.825 GB |
| *Adapting four backbones* | | | | | |
| Fine-Tuning | 100.00% | 100.00% | 21.675 GB | 100.00% | 1.738 GB |
| Multi-Adapter+ | 0.38% | 65.67% | 13.246 GB | 100.33% | 1.742 GB |
| ComBo adapter | 1.71% | 34.41% | 3.212 GB | 101.60% | 3.159 GB |

## 4.6 Computational efficiency

ComBo offers several computational benefits over existing baselines, particularly when combining multiple FMs. To compress multiple models into one, distillation methods require expensive training (e.g., SAM-CLIP and RADIOv2.5 used 64 large GPUs, with the latter training on 614M examples [4, 5]) and lack flexibility to incorporate new models without retraining. In contrast, ComBo can be trained in ∼10 minutes to probe four models on a VTAB-1k task using a consumer GPU.

Tab. 7 details computational costs of different adaptation methods for ViT-B/16 backbones. ComBo requires significantly fewer training FLOPs due to the absence of backpropagation through backbones: only 35.78% vs. 65.58% for Adapter+ (single backbone), and 34.41% vs. 65.67% for Multi-Adapter+ (four backbones). Memory requirements are also substantially lower, with ComBo using ∼3 GB vs. 13 GB and 22 GB for Multi-Adapter+ and fine-tuning. For instance, using an NVIDIA RTX 3090 Ti (24GB), ComBo can probe large models like DINOv2-g/14 that would exceed memory limits when adapted with Adapter+ without additional memory optimisation techniques. While ComBo introduces modest inference overhead due to its additional parameters and transformer processing, the impact is minimal in practice, adding at most 3.00% increase in FLOPs to the original backbone(s).

## 5 Conclusion

Motivated by the important role of foundation models in computer vision and by the differences between representations learned by existing models, we propose ComBo, a probing-based adapter designed to process feature maps from multiple frozen models. ComBo outperforms prior probing-based adapters without requiring task-specific hyperparameter tuning and demonstrates the ability to effectively combine features from multiple models on the VTAB-1k benchmark. Furthermore, we introduce a method using ComBo to efficiently estimate the task-relevance of different models. This enables practitioners to improve both efficiency and performance by focusing computational resources on the most relevant models. While ComBo alone does not yet match the performance of tuning-based approaches such as Adapter+, we show that it can successfully integrate the representations of multiple models that have been fine-tuned individually, offering a promising solution for efficient model combination.

**Limitations** Our work has several limitations. First, ComBo assumes consistent feature map shapes across models. While we use interpolation to handle heterogeneous architectures, this may degrade representation fidelity. Second, using a single affine projection for all layers may limit scalability as we want to include more model layers. More flexible or hierarchical projection mechanisms could improve scalability and representational capacity. Third, ComBo inherits representational biases from frozen models, which may affect downstream predictions. Fourth, while our scoring mechanism successfully ranks models by task-relevance, it remains unclear how to automatically determine the optimal number of models to maximise performance without evaluating each possible option. Finally, our experiments focus on ViT-B architectures and VTAB-1k's low-data classification tasks. Evaluation on larger models, higher-resource settings, and dense prediction tasks would help to establish the broader generalisability of ComBo.

## Acknowledgements

This work was supported by EPSRC Programme Grant "From Sensing to Collaboration" (EP/V000748/1), the EPSRC Centre for Doctoral Training in Autonomous Intelligent Machines and Systems (EP/S024050/1), Oxa, and the Digital Research Alliance of Canada.

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

# A Additional details

## A.1 Additional dataset details

We summarise the details of the datasets used in the VTAB benchmark [10] in Tab. 8.

Table 8: Details of the datasets in the VTAB benchmark [10] used. When experimenting on the VTAB-1k setting or estimating models' task-relevance, we use 800 training samples and 200 validation samples. When running our final tests on the VTAB-1k setting, we use the 1000 training samples.

| | | 1k setting | | full setting | | |
|---|---|---|---|---|---|---|
| Dataset | #Classes | Train | Val | Train | Val | Test |
| *Natural* | | | | | | |
| Caltech101 [38] | 102 | 800/1,000 | 200 | 2,754 | 306 | 6,084 |
| CIFAR-100 [39] | 100 | 800/1,000 | 200 | 45,000 | 5,000 | 10,000 |
| DTD [40] | 47 | 800/1,000 | 200 | 1,880 | 1,880 | 1,880 |
| Flowers102 [41] | 102 | 800/1,000 | 200 | 1,020 | 1,020 | 6,149 |
| Pets [42] | 37 | 800/1,000 | 200 | 2,944 | 736 | 3,669 |
| Sun397 [43] | 397 | 800/1,000 | 200 | 76,127 | 10,875 | 21,750 |
| SVHN [44] | 10 | 800/1,000 | 200 | 65,931 | 7,326 | 26,032 |
| *Specialised* | | | | | | |
| Patch Camelyon [45] | 2 | 800/1,000 | 200 | 262,144 | 32,768 | 32,768 |
| EuroSAT [46] | 10 | 800/1,000 | 200 | 16,200 | 5,400 | 5,400 |
| Resisc45 [47] | 45 | 800/1,000 | 200 | 18,900 | 6,300 | 6,300 |
| Retinopathy [48] | 5 | 800/1,000 | 200 | 35,126 | 10,906 | 42,670 |
| *Structured* | | | | | | |
| Clevr/count [8] | 8 | 800/1,000 | 200 | 63,000 | 7,000 | 15,000 |
| Clevr/distance [8] | 6 | 800/1,000 | 200 | 63,000 | 7,000 | 15,000 |
| DMLab [49] | 6 | 800/1,000 | 200 | 65,550 | 22,628 | 22,735 |
| dSprites/location [50] | 16 | 800/1,000 | 200 | 589,824 | 73,728 | 73,728 |
| dSprites/orientation [50] | 16 | 800/1,000 | 200 | 589,824 | 73,728 | 73,728 |
| KITTI/distance [51] | 4 | 800/1,000 | 200 | 6,347 | 423 | 711 |
| SmallNORB/azimuth [52] | 18 | 800/1,000 | 200 | 24,300 | 12,150 | 12,150 |
| SmallNORB/elevation [52] | 18 | 800/1,000 | 200 | 24,300 | 12,150 | 12,150 |

## A.2 Additional backbones details

We share details of all backbones used in our experiments in Tab. 9.

For ComBo we do not use `cls` or register tokens. For other baselines, the pooling used before the linear head depends on the model and is specified in the table.

Table 9: Details of backbones used in our experiments. We only report pooling strategies when they were used in our experiments, such as when tuning the model with Adapter+. All models are loaded and adapted from timm [35], except for RADIOv2.5 which is loaded from the PyTorch Hub. The "-CPE" suffix indicates the use of Cropped Position Embeddings as implemented in RADIOv2.5 [5].

| Model | Architecture | Model path in timm or PyTorch Hub | Pooling used |
|---|---|---|---|
| CLIP [1] | ViT-B/16 | timm/vit_base_patch16_clip_224.openai | - |
| DFN CLIP [53] | ViT-B/16 | timm/vit_base_patch16_clip_224.dfn2b | `cls` token |
| DINOv2 [6] | ViT-B/14 | timm/vit_base_patch14_reg4_dinov2.lvd142m | `cls` token |
| ImageNet-21K [7] | ViT-B/16 | timm/vit_base_patch16_224.orig_in21k | `cls` token |
| MAE [2] | ViT-B/16 | timm/vit_base_patch16_224.mae | - |
| RADIOv2.5 [5] | ViT-B/16-CPE | NVlabs/RADIO/radio_v2.5-b | summary (equivalent to `cls` token) |
| SAM [3] | ViT-B/16 | timm/samvit_base_patch16.sa1b | average pooling |
| SigLIP [36] | ViT-B/16 | timm/vit_base_patch16_siglip_224.webli | average pooling |

### A.3 Additional training details

**Computational details** All our experiments are performed using BFloat16 Mixed precision. Our experiments were mainly performed on a machine with one NVIDIA GeForce RTX 3090 Ti GPU and an Intel(R) Xeon(R) Silver 4210R CPU @ 2.40GHz with 40 logical CPUs. With our system, training one ComBo adapter probing one backbone on one VTAB-1k dataset takes approximately 5 minutes, and training one ComBo adapter probing four backbones on one VTAB-1k dataset takes approximately 10 minutes.

**Tuning and probing baselines** Results presented in Tab. 1 are reported from Adapter+ [32] for the tuning methods and the linear probing result, and from SMP [12] for Head2toe and SMP.

**Linear probing different layers of FMs** Results of Fig. 1a and Fig. 5 are produced using the following procedure: (1) average-pooling all tokens at the output of the selected block, (2) applying a LayerNorm layer, and (3) learning a linear classifier on top of the pooled output. We train for 100 steps with a batch size of 128 using the AdamW optimizer [34]. We use no weight decay and no learning rate scheduler. The learning rate is selected based on validation performance from the set $[1.0, 0.1, 0.01, 0.001, 0.0001, 0.00001]$.

**Training Adapter+ and Multi-Adapter+** For all experiments involving Adapter+ applied to foundation models, we use the implementation from `https://github.com/visinf/adapter_plus`. We follow the recommended settings from the original paper [32], which include post-adapter placement, channel-wise scaling, and Houlsby initialization. We use a rank of 16 for our experiments as it led to the highest Global Average. We train the adapter with the same optimisation settings and schedule as the original paper and our method, discussed in Sec. 4. Following the authors' recommendations, we apply stochastic depth during training. For the frozen network, we use linearly increasing drop rates from 0 to 0.1 as a function of network depth. For the adapters, we use a fixed drop rate of 0.1.

For Multi-Adapter+, we first apply Adapter+ blocks to all backbones. We then pool their outputs using model-specific strategies (see Sec. A.2 for details). Finally, we stack the resulting pooled features before applying a linear classifier.

### A.4 Additional details about the generation of images that maximise FMs activations

Images in Figs. 2 and 4 were generated using the method from Ghiasi et al. [13]. We used the implementation available at `https://github.com/anonymous2022icml/ViTViS`. Neurons maximised to produce images are randomly selected for each layer and model. We used 400 optimization steps with a total variation coefficient of 0.1 and strength $\lambda_{tv} = 0.0005$. The learning rate was set to 1.0 for CLIP and MAE, and 0.1 for other models.

### A.5 Computational requirements computation

The computational cost comparison in Tab. 7 is based on a batch size of 32. These measurements were obtained when adapting to the dSprites/location dataset from the VTAB-1k benchmark. This dataset contains 16 classes, which affects the number of parameters in the classifier head. However, the head represents only a small fraction of the compute load for full fine-tuning, Adapter+, and ComBo. Therefore, the relative differences between adaptation techniques would remain similar across all VTAB benchmark tasks, regardless of the number of classes.

The results for adapting one backbone are based on adapting the ViT-B/16 model trained on ImageNet-21K. The results for adapting four backbones are based on adapting the DFN CLIP, DINOv2, SAM, and SigLIP ViT-B models.

### A.6 Image and icons licenses

The image of a space probe used in Fig. 3 is from work performed by ATG under contract for ESA and is shared under a CC BY-SA IGO 3.0 license. Icons used in our figures come from Font Awesome and are shared under a CC BY 4.0 license.

# B Additional results

## B.1 Additional visualisations of images that maximise neuron activations

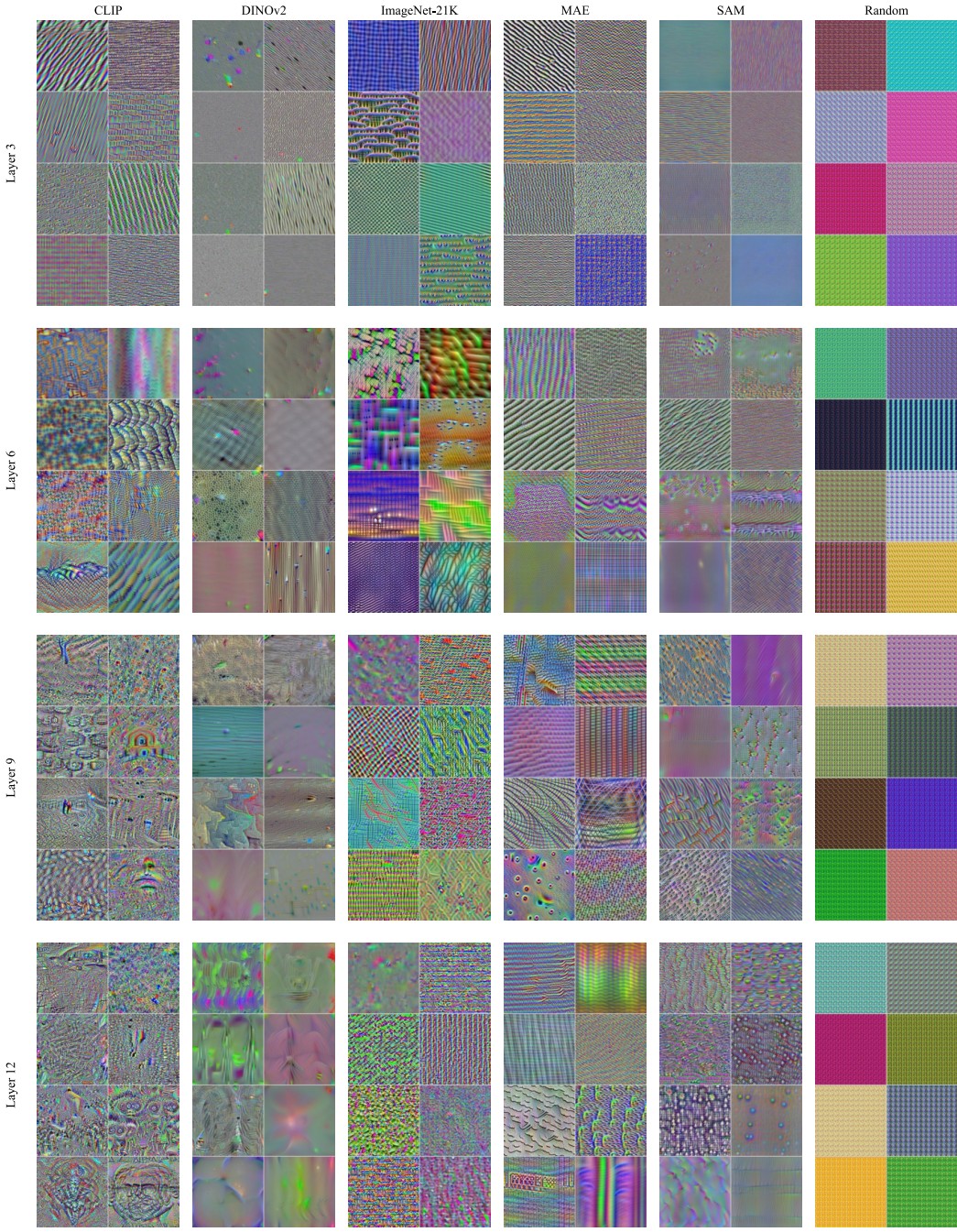

Figure 4: Additional visualisations extending Fig. 2. These images are optimised to maximise activations of different neurons across different layers using the technique from Ghiasi et al. [13]. We also include visualisations for a ViT with randomly initialised weights. All images are generated from neurons that were randomly sampled.

In Fig. 4, we present additional images from the experiment of Fig. 2 that maximise different neurons of different layers of FM. We include an additional "Random" model with a ViT-B/16 architecture and randomly initialised weights to validate that the observed patterns come from learned representations.

Our images show that different layers of different models tend to have different representations, and the most relevant layer to gather features for a downstream task is task-dependent. This finding validates the value of probing architectures that can simultaneously access features from all layers of multiple models, such as ComBo.

## B.2 Additional results from probing different layers of different foundation models on VTAB-1k

In Fig. 5, we also provide extended results of our linear probing experiment presented in Fig. 1a, with all datasets of the VTAB-1k benchmark.

Our observations hold, with the best-performing layer and model depending on the dataset. For natural tasks such as Caltech101, CIFAR-100, DTD, Flowers102, Pets, and Sun397, we can see that models trained with more semantic-based supervision such as CLIP, DINOv2, and ImageNet-21K tend to outperform MAE and SAM, usually with features from the penultimate or last layer. However, this pattern does not hold for different tasks, with intermediate layers being sometimes better suited (e.g., SVHN) or MAE performing best (e.g., Clevr/distance).

## B.3 Training adapters with more training samples

To evaluate how different adaptation methods scale with training data size, we conducted experiments using the full setting of the VTAB benchmark (which provides access to complete training datasets, as opposed to the few-shot setting) and present results in Fig. 6.

For this experiment, we randomly sampled 1000, 5000, 10000, and 20000 images from each dataset's training split and trained different adapters on these subsets to examine how the performance of ComBo and other methods scales with increasing training data. For linear probing, we applied it to the `cls` token representation after the final transformer layer.

The results demonstrate that ComBo shows consistent performance improvements as more training data becomes available across all tested datasets. Notably, on the SmallNORB/azimuth dataset, ComBo surpasses Adapter+ performance at higher sample counts. ComBo's ability to benefit from more training data can be attributed to its transformer-based approach to processing different tokens, which naturally benefits from larger training datasets.

Interestingly, on the Sun397 dataset, linear probing initially outperforms ComBo, though this performance gap narrows with additional training data. This observation highlights the effectiveness of the regularization inherent in training only a linear classifier on top of the most relevant pre-trained features, which aligns with our earlier findings from the linear probing experiments on Sun397 (Fig. 5g). However, the diminishing gap suggests that ComBo's more flexible architecture increasingly leverages additional data to close this performance difference.

## B.4 Training adapters with more classes

We also validate the behaviour of ComBo on a classification task with various number of classes. To do this, we use the dataset from the 2018 FGVCx Fungi Classification Challenge[54] which contains images of fungi with 1394 species to classify.

We experiment with various number of classes by randomly selecting different numbers of subsets of classes, and evaluating adaptation results on the resulting dataset. We consider 5, 10, 20, 50, 100, 250, 500, 750, 1000, and 1394 classes. For each number of class, we randomly select classes to keep and use all available data for these classes. This is done over three seeds for each number of classes, leading to different subsets of classes and data for each run.

We present results in Fig. 7, where all methods display the same behaviour. As more classes are included, the task becomes more difficult and the performance of all methods drops. Linear probing in particular suffers more from the increase in number of classes.

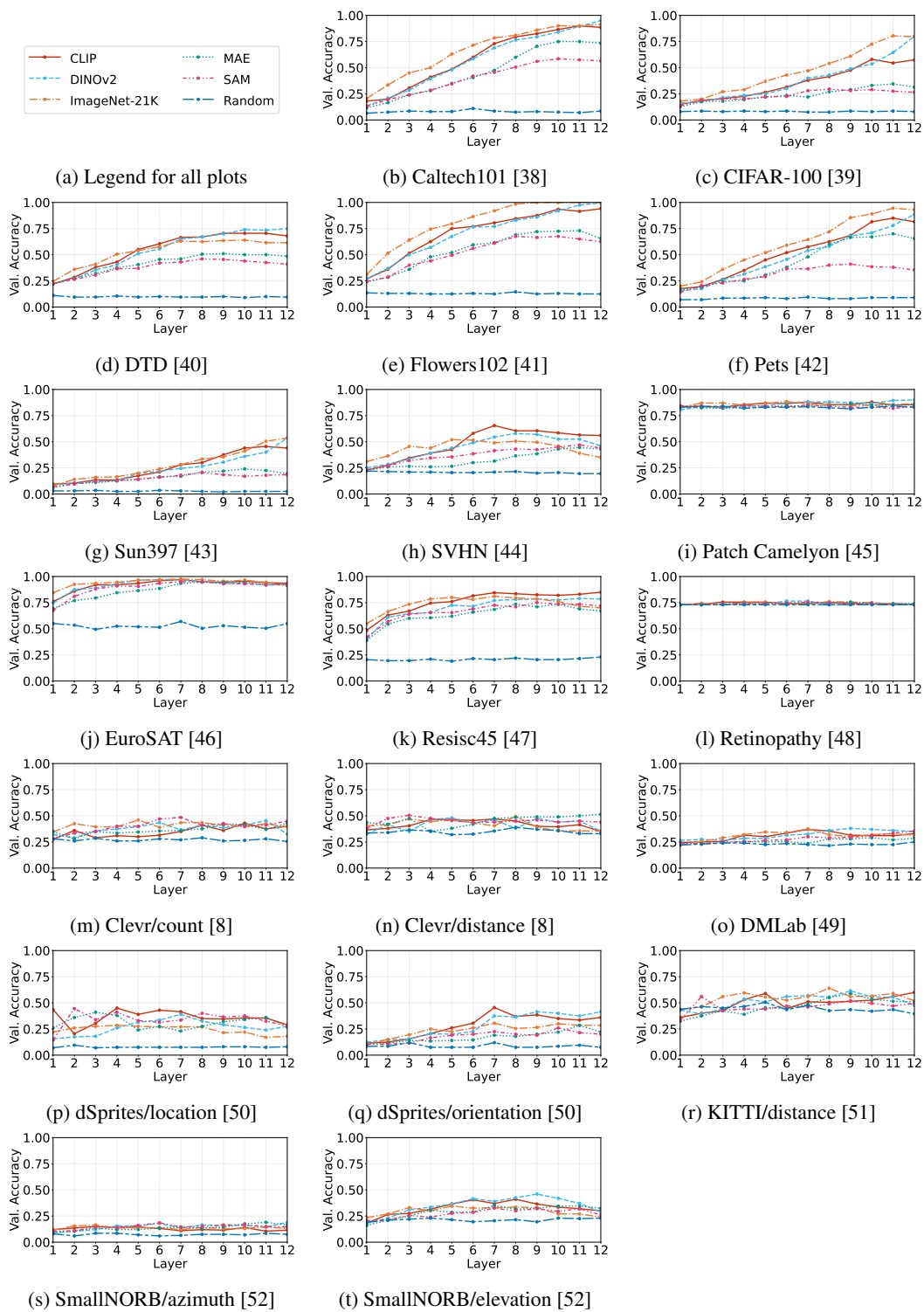

Figure 5: Results of linear probing as presented in Fig. 1a for all VTAB-1k [10] datasets. The plots present validation accuracies of linear probing on activations from different layers of ViTs trained from CLIP [1], DINOv2 [6], ImageNet-21K [7], MAE [2], SAM [3], as well as a randomly initialised ViT.

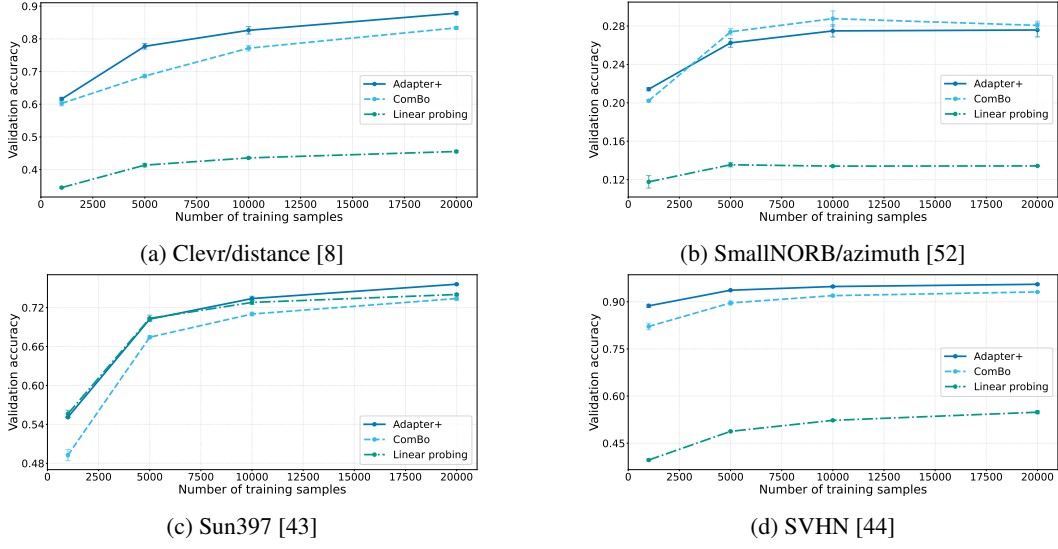

(a) Clevr/distance [8]

(b) SmallNORB/azimuth [52]

(c) Sun397 [43]

(d) SVHN [44]

Figure 6: Results of using various adaptation methods with a ViT-B/16 trained on ImageNet-21K [7] for datasets from the VTAB benchmark [10] with increasing number of training samples. Results are reported on the full validation set. The results are averaged over 3 runs, with error bars showing standard deviation.

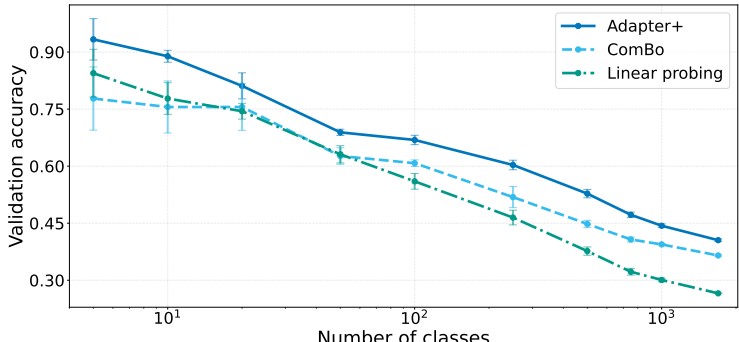

Figure 7: Results of using various adaptation methods with a ViT-B/16 trained on ImageNet-21K [7] on the FGVCx Fungi Classification Challenge dataset [54] with increasing number of classes to classify. Results are reported on the validation set. The sampling of subsets of classes is done over three seeds, and our plots display average accuracy with error bars showing standard deviation.

## B.5 Additional ablations of ComBo

We also present additional ablation results in Tab. 10 to validate our design choices.

**Keeping the cls token**   In our ComBo experiment, we decide not to use the `cls` token or register tokens to simplify the design choices when combining multiple models which might have different variations of these tokens. Moreover, we expect our transformer to be able to learn attention to relevant information needed for the task with its own `cls` token, compensating for the absence of any original global token from the pre-trained backbone. Our ablation shows a minimal improvement (74.1% vs 73.8% for the Global Average) when keeping the original `cls` token.

When probing multiple models, different strategies to handle the `cls` token are possible, such as assigning them to new compressed tokens to feed to our transformer, but we have not explored this thoroughly as we find the use of tokens associated to original image patches more general and simple.

**Taking activations elsewhere**   Our experiments also rely on probing activations from the output of each block. However, previous probing methods [11, 12] use multiple locations within each block to extract features. We find that ComBo's performance is competitive without the need to add

Table 10: **Validation of ComBo design choices**. We report accuracy (%) on the *val* set when adapting the ViT-B/16 model trained on ImageNet-21K.

| | Natural | Specialised | Structured | Global Average |
|---|---|---|---|---|
| ComBo adapter | 79.3 | 85.6 | 56.5 | 73.8 |
| *Probed features (default: no `cls` or register tokens, probing the output of all blocks)* | | | | |
| Keeping the `cls` token | 79.5 | 86.1 | 56.5 | 74.1 |
| Probing after the second normalisation layer in each block | 78.4 | 86.1 | 53.5 | 72.7 |
| Probing only the output of the first 6 blocks | 56.5 | 83.3 | 55.2 | 65.0 |
| Probing only the output of the last 6 blocks | 77.6 | 82.5 | 47.3 | 69.1 |
| Probing the outputs of only blocks 2, 4, 6, 8, 10, and 12 | 78.9 | 85.5 | 56.2 | 73.5 |
| *Different ComBo transformer settings (default: 6 blocks, embedding dimension of 128, 2 heads)* | | | | |
| With a depth of 1 block | 79.3 | 84.5 | 53.9 | 72.6 |
| With a depth of 2 blocks | 79.2 | 86.0 | 56.1 | 73.8 |
| With a depth of 12 blocks | 78.8 | 85.5 | 55.5 | 73.2 |
| With an embedding dimension of 64 and 1 head | 78.1 | 85.5 | 55.8 | 73.1 |
| With an embedding dimension of 256 and 4 heads | 79.6 | 86.1 | 53.3 | 73.0 |
| *Different classification head (default: linear)* | | | | |
| With a Gaussian Naive Bayes classifier head | 77.3 | 69.8 | 54.8 | 67.3 |

additional locations, and prefer to keep fewer locations within each block to maintain a lower number of parameters of our affine projection.

In our ablation, we validate that the output of the block is a good choice to collect activations by comparing it to using the output of the second normalisation layer in each block. This could be a suitable location as features are already normalised, but the performance we obtain is slightly lower (72.7% vs 73.8% for the Global Average), indicating that the output of the block might have more informative activations.

**Probing different combinations of layers** We also validate the importance of probing all layers rather than only specific ones. We find that only probing the first 6 blocks leads to a significant drop in performance (65.0% vs 73.8% for the Global Average), with the largest difference appearing in the Natural tasks (56.5% vs 79.3%), which usually rely on more semantic features located in late layers. On the other hand, probing only the last 6 blocks also leads to a drop in performance (69.1% vs 73.8% for the Global Average), but with the largest difference appearing in the Structured tasks (47.3% vs 56.5%). Using 6 layers better spread across the network (2, 4, 6, 8, 10, and 12) leads to a much stronger performance (73.5% for the Global Average), but still does not reach the performance of probing all layers (73.8%).

**Different transformer sizes** The architecture choice of our transformer is also validated in Tab. 10, with different embedding dimensions or number of blocks leading to drops in performance.

We expect that this is related to the small amount of training data in the VTAB-1k setting and to the complexity of the tasks. We believe that more complex tasks with more data could benefit from larger transformer architectures within ComBo.

**Different classification head** In FiT [55], Shysheya et al. observed that using a Naive Bayes final layer classifier performed better than a linear classifier in a few-shot setting. We also experiment by using the same head as the FiT-LDA variant, but find that the linear head still outperforms it in our case, with a significant difference in the specialised domain. However, we note that we did not experiment with using an episodic fine-tuning setting, which the authors note is crucial to obtain the best results with the Naive Bayes head.

## B.6 Removing individual models when probing of multiple foundation models

In Tab. 11, we experiment with probing multiple FMs and removing some of them to see how the performance of ComBo is affected. As observed in Tab. 2, SAM performed much worse than other models considered when probed. This is reflected in the best performance of ComBo when removing SAM features (76.9% vs 76.2% for the Global Average when probing all models). On the other hand, DINOv2 is the best-performing model overall when probed individually (Tab. 2), and removing its

Table 11: **Performance when probing different combinations of FMs.** We report accuracy (%) on the *val* set. We compare probing all four FMs considered with ComBo (DFN CLIP [53], DINOv2 [6], SAM [3], and SigLIP) [36], and removing each one from the set of probed models. Best results are in **bold** and second-best are underlined.

| | Natural | | | | | | | | Specialised | | | | | Structured | | | | | | | | | Global Average |
|---|---|---|---|---|---|---|---|---|---|---|---|---|---|---|---|---|---|---|---|---|---|---|---|
| | Caltech101 | CIFAR-100 | DTD | Flowers102 | Pets | Sun397 | SVHN | *Average* | Camelyon | EuroSAT | Resisc45 | Retinopathy | *Average* | Clevr-Count | Clevr-Dist | DMLab | dSpr-Loc | dSpr-Ori | KITTI-Dist | sNORB-Azim | sNORB-Elev | *Average* | |
| All 4 FMs | 97.2 | 76.0 | 73.3 | **100.0** | 93.2 | 46.2 | **87.0** | 81.8 | 87.2 | _97.3_ | 89.8 | _71.2_ | 86.4 | 89.7 | **64.5** | _49.0_ | **92.8** | 48.0 | 72.2 | 18.7 | _47.7_ | 60.3 | _76.2_ |
| *Removing individual backbones from probing* | | | | | | | | | | | | | | | | | | | | | | | |
| w/o DFN CLIP | **97.8** | 74.3 | _74.2_ | _99.8_ | **94.0** | _48.3_ | 85.8 | _82.1_ | **88.0** | 97.0 | **90.5** | 69.8 | 86.3 | **91.8** | 63.0 | 48.8 | 91.2 | 47.3 | 70.7 | _22.2_ | 45.3 | 60.0 | 76.1 |
| w/o DINOv2 | 97.0 | 67.2 | 72.3 | 99.3 | _93.5_ | 47.7 | 82.7 | 80.0 | 84.8 | 96.8 | 90.0 | _72.8_ | 86.1 | 90.0 | _63.7_ | 46.8 | 90.2 | **50.3** | **81.8** | 20.5 | 42.8 | _60.8_ | 75.6 |
| w/o SAM | 97.2 | _76.3_ | **75.5** | _99.8_ | 93.0 | **49.8** | 86.5 | **82.6** | _87.7_ | **97.8** | _90.2_ | **72.8** | **87.1** | _90.7_ | 63.2 | **52.2** | _91.8_ | 49.3 | 70.8 | 21.2 | **49.2** | **61.0** | **76.9** |
| w/o SigLIP | _97.3_ | **76.7** | 70.8 | _99.8_ | 91.3 | 44.0 | _86.7_ | 81.0 | _87.7_ | 96.5 | 89.8 | 70.0 | 86.0 | 89.8 | 63.0 | 48.3 | 91.3 | _50.0_ | _74.5_ | **22.5** | 43.3 | 60.3 | 75.8 |

features from the probing leads to the highest drop in performance (75.6% vs 76.2% for the Global Average).

## B.7  Detailed models' task-relevance estimation results

Tab. 12 presents the detailed importance scores generated by our method for each model on each VTAB-1k task. These scores reflect each model's estimated task-relevance and are used to select which models to probe for each dataset. In Tab. 2 and Tab. 5, we demonstrate how these importance scores guide model selection, with Tab. 2 specifically showing results when selecting the top two models (those with the highest importance scores) for each task.

The importance scores align with the observations from Sec. B.6. DINOv2 consistently achieves the highest or second-highest importance scores across most tasks, reflecting its strong performance on VTAB-1k. Conversely, SAM generally receives lower importance scores, corresponding to its weaker performance on these visual recognition tasks.

Notably, the optimal number of models varies substantially across tasks, as shown in the bottom row of Tab. 12. Even within a single category, the best ensemble size ranges from a single model to all four models, highlighting the importance of task-specific model selection.

Table 12: **Estimated importance scores for each model on each VTAB-1k task.** Scores are computed using our proposed method and averaged over three training seeds. The bottom row shows the optimal number of models (selected by highest importance scores) that achieved the best performance on each task. Highest scores are in **bold** and second highest are underlined.

| | Natural | | | | | | | | Specialised | | | | | Structured | | | | | | | | | Global Average |
|---|---|---|---|---|---|---|---|---|---|---|---|---|---|---|---|---|---|---|---|---|---|---|---|
| | Caltech101 | CIFAR-100 | DTD | Flowers102 | Pets | Sun397 | SVHN | *Average* | Camelyon | EuroSAT | Resisc45 | Retinopathy | *Average* | Clevr-Count | Clevr-Dist | DMLab | dSpr-Loc | dSpr-Ori | KITTI-Dist | sNORB-Azim | sNORB-Elev | *Average* | |
| DFN CLIP | 2.91 | _3.61_ | _2.70_ | _3.21_ | 5.92 | 3.97 | _1.29_ | _3.37_ | 2.98 | 0.40 | 6.50 | **3.86** | _3.44_ | 2.50 | **6.74** | _6.43_ | **4.56** | _5.55_ | 2.80 | _2.50_ | 4.37 | _4.43_ | _3.75_ |
| DINOv2 | **4.51** | **6.20** | **4.09** | **4.72** | _6.27_ | **5.08** | **3.38** | **4.89** | **3.52** | **2.07** | _6.51_ | _3.62_ | **3.93** | **3.13** | 6.14 | **6.65** | 2.80 | **6.70** | **3.45** | **4.94** | **7.15** | **5.12** | **4.65** |
| SAM | 2.36 | 2.09 | 2.03 | 2.44 | 4.92 | 1.82 | 0.56 | 2.32 | 2.11 | 0.32 | 4.98 | 3.19 | 2.65 | 2.20 | 5.26 | 5.85 | _3.48_ | 4.92 | 2.78 | 2.10 | 3.14 | 3.72 | 2.89 |
| SigLIP | _2.93_ | 2.55 | 2.48 | 3.05 | **6.52** | _4.22_ | 0.35 | 3.16 | 2.00 | 0.15 | 6.02 | 2.79 | 2.74 | _2.66_ | _6.26_ | 5.01 | 3.11 | 4.67 | 1.97 | 1.30 | 3.56 | 3.57 | 3.15 |
| Best # of models | 1 | 2 | 3 | 1 | 1 | 2 | 4 | 2 | 2 | 1 | 3 | 2 | 2 | 2 | 4 | 2 | 4 | 1 | 1 | 2 | 1 | 2.125 | 2.04 |

## B.8  Standard deviations for test result tables

Finally, we report the standard deviations for all test results presented in the main paper in Tabs. 13 to 15.

Table 13: Standard deviations (%) over the three seeds for produced results in Tab. 1. For the average columns, we report the average of the standard deviations across all datasets in each category.

| | Natural | | | | | | | | Specialised | | | | | Structured | | | | | | | | | |
| | Caltech101 [38] | CIFAR-100 [39] | DTD [40] | Flowers102 [41] | Pets [42] | Sun397 [43] | SVHN [44] | *Average* | Camelyon [45] | EuroSAT [46] | Resisc45 [47] | Retinopathy [48] | *Average* | Clevr-Count [8] | Clevr-Dist [8] | DMLab [49] | dSpr-Loc [50] | dSpr-Ori [50] | KITTI-Dist [51] | sNORB-Azim [52] | sNORB-Elev [52] | *Average* | **Global Average** |
|---|---|---|---|---|---|---|---|---|---|---|---|---|---|---|---|---|---|---|---|---|---|---|---|
| ComBo (ours) | 0.40 | 0.26 | 0.28 | 0.03 | 0.47 | 0.18 | 0.74 | 0.34 | 0.41 | 0.27 | 0.21 | 0.26 | 0.29 | 0.11 | 0.25 | 0.55 | 1.63 | 0.10 | 0.88 | 0.37 | 1.33 | 0.65 | 0.43 |

Table 14: Standard deviations (%) over the three seeds for produced results in Tab. 2. For the average columns, we report the average of the standard deviations across all datasets in each category.

| | Natural | | | | | | | | Specialised | | | | | Structured | | | | | | | | | |
| | Caltech101 | CIFAR-100 | DTD | Flowers102 | Pets | Sun397 | SVHN | *Average* | Camelyon | EuroSAT | Resisc45 | Retinopathy | *Average* | Clevr-Count | Clevr-Dist | DMLab | dSpr-Loc | dSpr-Ori | KITTI-Dist | sNORB-Azim | sNORB-Elev | *Average* | **Global Average** |
|---|---|---|---|---|---|---|---|---|---|---|---|---|---|---|---|---|---|---|---|---|---|---|---|
| *Probing individual foundation models with ComBo (ours)* | | | | | | | | | | | | | | | | | | | | | | | |
| DFN CLIP [53] | 0.60 | 0.42 | 0.45 | 0.13 | 0.27 | 0.19 | 0.80 | 0.41 | 0.96 | 0.14 | 0.05 | 0.32 | 0.37 | 0.51 | 0.38 | 0.24 | 1.08 | 0.25 | 0.75 | 1.19 | 0.15 | 0.57 | 0.45 |
| DINOv2 [6] | 0.13 | 0.08 | 0.18 | 0.02 | 0.35 | 0.40 | 0.68 | 0.26 | 0.44 | 0.18 | 0.58 | 0.51 | 0.43 | 0.52 | 0.80 | 0.97 | 5.11 | 0.44 | 0.43 | 0.63 | 1.37 | 1.29 | 0.66 |
| SAM [3] | 0.74 | 0.92 | 0.64 | 0.57 | 0.38 | 0.18 | 0.63 | 0.58 | 0.30 | 0.42 | 0.76 | 0.35 | 0.46 | 0.18 | 0.35 | 0.76 | 2.29 | 0.21 | 0.99 | 1.58 | 0.44 | 0.85 | 0.63 |
| SigLIP [36] | 0.50 | 0.39 | 0.16 | 0.09 | 0.33 | 0.61 | 0.24 | 0.33 | 1.10 | 0.26 | 0.25 | 0.54 | 0.54 | 1.45 | 0.19 | 0.47 | 1.06 | 0.24 | 0.96 | 0.04 | 1.36 | 0.72 | 0.53 |
| *Probing all four foundation models* | | | | | | | | | | | | | | | | | | | | | | | |
| ComBo (ours) | 0.23 | 0.48 | 0.22 | 0.02 | 0.33 | 0.44 | 0.28 | 0.29 | 0.81 | 0.16 | 0.61 | 0.41 | 0.50 | 0.90 | 0.25 | 0.62 | 1.83 | 0.61 | 1.90 | 2.77 | 0.90 | 1.22 | 0.67 |
| *Distilling all four foundation models above and tuning the resulting model with Adapter+ [32]* | | | | | | | | | | | | | | | | | | | | | | | |
| RADIOv2.5 [5] | 0.89 | 0.35 | 0.28 | 0.28 | 0.19 | 0.31 | 0.17 | 0.35 | 0.21 | 0.32 | 0.44 | 0.31 | 0.32 | 0.30 | 0.37 | 0.51 | 0.16 | 0.65 | 0.65 | 0.32 | 1.70 | 0.58 | 0.42 |

Table 15: Standard deviations (%) over the three seeds for produced results in Tab. 3. For the average columns, we report the average of the standard deviations across all datasets in each category.

| | Natural | | | | | | | | Specialised | | | | | Structured | | | | | | | | | |
| | Caltech101 | CIFAR-100 | DTD | Flowers102 | Pets | Sun397 | SVHN | *Average* | Camelyon | EuroSAT | Resisc45 | Retinopathy | *Average* | Clevr-Count | Clevr-Dist | DMLab | dSpr-Loc | dSpr-Ori | KITTI-Dist | sNORB-Azim | sNORB-Elev | *Average* | **Global Average** |
|---|---|---|---|---|---|---|---|---|---|---|---|---|---|---|---|---|---|---|---|---|---|---|---|
| *Individual adapted models* | | | | | | | | | | | | | | | | | | | | | | | |
| DFN CLIP [53] | 0.11 | 0.39 | 0.11 | 0.11 | 0.55 | 0.05 | 0.30 | 0.23 | 0.50 | 0.10 | 0.10 | 0.11 | 0.20 | 0.41 | 0.53 | 0.29 | 0.20 | 0.30 | 0.76 | 0.42 | 1.59 | 0.56 | 0.33 |
| DINOv2 [6] | 0.23 | 0.21 | 0.07 | 0.02 | 0.06 | 0.06 | 0.35 | 0.14 | 0.41 | 0.15 | 0.28 | 0.35 | 0.29 | 1.26 | 0.34 | 0.21 | 0.57 | 0.59 | 0.63 | 0.97 | 3.00 | 0.95 | 0.46 |
| SAM [3] | 0.45 | 0.37 | 0.76 | 0.32 | 1.51 | 0.27 | 0.45 | 0.59 | 0.37 | 0.16 | 0.35 | 0.29 | 0.29 | 0.92 | 0.50 | 0.70 | 0.72 | 1.26 | 1.26 | 0.42 | 0.24 | 0.75 | 0.54 |
| SigLIP [36] | 0.42 | 0.52 | 0.29 | 0.18 | 0.24 | 0.43 | 0.13 | 0.31 | 0.17 | 0.14 | 0.15 | 0.55 | 0.25 | 0.30 | 0.04 | 0.18 | 0.35 | 0.25 | 0.40 | 0.22 | 0.53 | 0.28 | 0.28 |
| *Approaches combining models above* | | | | | | | | | | | | | | | | | | | | | | | |
| Multi-Adapter+ | 0.02 | 0.04 | 0.11 | 0.02 | 0.35 | 0.14 | 0.44 | 0.16 | 0.51 | 0.17 | 0.05 | 0.61 | 0.34 | 5.40 | 0.61 | 0.53 | 1.06 | 0.47 | 1.21 | 0.23 | 0.96 | 1.31 | 0.60 |
| ComBo (ours) | 0.64 | 0.20 | 0.30 | 0.10 | 0.40 | 0.35 | 0.07 | 0.30 | 0.10 | 0.06 | 0.28 | 1.77 | 0.55 | 0.87 | 0.25 | 1.32 | 0.14 | 0.71 | 0.57 | 1.88 | 0.07 | 0.73 | 0.53 |

