# OpenReview forum: "Fantastic Features and Where to Find Them: A Probing Method to combine Features from Multiple Foundation Models"
_NeurIPS.cc/2025/Conference — NeurIPS 2025 poster_

### Official Review · Reviewer_VFAP · 2025-06-14

**Clarity:** 3
**Significance:** 3
**Originality:** 2
**Rating:** 4
**Confidence:** 3

**Summary:**

The manuscript builds on an empirical observation that contemporary foundation models learn different representations (Fig. 2). Consequently, the manuscript proposes a probing method which combines features from multiple frozen foundation models into a single representation. The method begins by taking activations from all layers across all models.  These activations are first processed to equalise the token count for all models. This is done by reshaping tokens from each layer into a 2D grid and applying bilinear interpolation. The tokens are then standardised and concatenated with their counterparts from other models. The resulting matrix S has a dimensionality of (token count * layer count) x (sum of all model dimensions). The matrix S is then affinely projected with learnable parameters (W, b) into a lower-dimensional space. The resulting matrix is then concatenated with a cls token and fed to a shallow transformer F that generates the final representation. Learnable parameters are the task head, transformer F, cls token, and the projection (W, b). Interestingly, the resulting procedure keeps foundation models frozen, which greatly reduces the computational requirements. Experimental evaluation on VTAB-1k indicates that the proposed method on average beats relevant previous works.

**Questions:**

Q1. It seems that the manuscript misses the most important experiment. Please answer W1 in detail.

Q2. What if one or more foundation models are irrelevant for some downstream tasks. Could ComBo detect this and exclude such a foundation model during the inference? The benefit would be in inference speedup.

Q3. The VTAB benchmark has a relatively low class count across the tasks (max <400). How does ComBo behave in setups with large class count (over 1k classes) and the VTAB full setting (Table 7)?

**Ethical Concerns:**

["NO or VERY MINOR ethics concerns only"]

**Final Justification:**

The manuscript proposes a probing method which combines features from multiple frozen foundation models into a single representation, which is interesting and potentially useful in practice. The author's response resolved my concerns, so I increased the score. Also, I thank the authors for a clear and elaborate response supported by sufficient experiments.

**Limitations:**

yes

**Paper Formatting Concerns:**

No concerns

**Quality:**

3

**Strengths And Weaknesses:**

Strenghts:

S1. The manuscript addresses an interesting problem.

S2. The proposed method is intuitive and easy to follow.

S3. The presented ablations are insightful.

S4. The proposed method performs well on different tasks with a constant hyperparameter setting.

Weaknesses:

W1. Table 1 shows that ComBo improves over SMP and Head2Toe when probing a single foundation model. However, a similar comparison is missing in Table 2 (merging features from multiple foundation models). It is hard to conclude on the effectiveness of the proposed method w.r.t related work without these crucial results.

W2. The method focuses solely on transformer architectures and does not comment on the integration of other architectures (e.g. convolutional models).


Typos:
- L144. Linear projection -> you mean affine projection (linear projection W + translation b)

---

> ### Author Rebuttal · Authors · 2025-07-31
>
> We thank the reviewer for their positive feedback on our intuitive method design, insightful ablations, and performance.
>
> **Overview**: We address your concerns about missing Head2Toe/SMP comparisons for multiple foundation models (Q1), clarify our approach to non-transformer architectures (W2), present new automatic model exclusion results (Q2), and provide scalability experiments (Q3).
>
> ## Replies to your concerns and questions
> ### 1. (Q1, W1) Lack of SMP and Head2Toe results in the setting with multiple foundation models
>
> **Evaluating SMP and Head2Toe with multiple backbones would be computationally prohibitive, while ComBo demonstrates superior performance with a single model.**
>
> We agree that ideally, we would compare against SMP and Head2Toe when probing multiple foundation models; however, we decided against this for practical reasons.
>
> First, both Head2Toe and SMP rely on extensive hyperparameter sweeps for each dataset, including learning rates, regularisation strengths, training duration, and target pooling size. Extending this to multiple foundation models would require expensive hyperparameter sweeps for SMP and Head2Toe on each dataset-backbone combination, which we cannot carry out due to computational limitations. Furthermore, SMP's implementation is not public, challenging reproducibility.
>
> In contrast, ComBo performs favourably against these methods with a single setting across 19 VTAB-1k tasks, without requiring hyperparameter sweeps. This makes ComBo more practical than existing approaches, and Table 1 clearly highlights ComBo's benefits over SMP and Head2Toe even in the single-model setting.
>
> Given ComBo's strong performance in Table 1, its practical advantages, and the substantial challenges of evaluating SMP and Head2Toe with multiple backbones, we believe the current validation is sufficient and have preferred using our resources to focus on additional experiments, ablations, and studies.
>
> ### 2. (W2) Integration of non-transformer architectures
> We thank the reviewer for highlighting this missing discussion. We agree this should be discussed in the paper and will update it. We provide architectural considerations here.
>
> **ViT-based approaches are standard**
>
> Whilst our evaluation focuses on limited architectures, we believe our findings remain useful. Our evaluation focuses on ViT-based architectures as they represent the dominant paradigm in modern foundation models. We expect ComBo to remain straightforward for most recent and future foundation models.
>
> **The general methodology of ComBo could be adapted to other architectures**
>
> The core principle of ComBo can be extended beyond this specific architecture family: The core methodology remains (1) extract features from multiple layers of multiple models, (2) align them spatially in an architecture-dependent manner, (3) compress them via a learned projection, (4) process the compressed features with a transformer. The spatial alignment step can be customised for different architectures while maintaining the overall framework.
>
> **Interpolation of feature maps is already used for different tokenisations and could apply to CNNs**
>
> We already handle different patch sizes, with DINOv2 using 14×14 patches compared to 16×16 for other models. We manage this through bilinear interpolation. This approach can likely extend to CNN feature maps by treating spatial locations as tokens.
>
>
> ### 3. (Q2) Novel automatic model exclusion method
>
> **Our backbone selection method reduces the average number of backbones from 4 to 1.42 whilst maintaining performance.**
>
> We thank the reviewer for this suggestion on ComBo detecting and excluding irrelevant models, also mentioned by other reviewers. Following this feedback, we developed and tested a new automatic model selection approach to address these concerns.
>
> Our two-stage method is inspired by Head2Toe [11], but instead of selecting features to retain, it selects backbones.
>
> **Stage 1 - Learning backbone importance**: We train ComBo with a novel regularisation term applied to projection weights Λ. For each backbone, we compute the L2 norm of all its projection weights across all layers to get an importance score. We sum importance scores from all backbones and add this as a regularisation term to minimise in our loss.
>
> **Stage 2 - Pruning and retraining**: After first-stage training, we measure resulting importance scores for each backbone. We prune backbones whose norms fall below 90% of the highest-scoring backbone, then retrain ComBo from scratch without regularisation using only selected backbones.
>
> ### Experimental results
> With our novel pruning mechanism:
> - **Performance is maintained**: ComBo with pruning achieves 78.1% vs 78.2% for full ComBo (only 0.1% drop)
> - **Efficiency is improved**: On average over the 19 datasets, we select 1.42 backbones per dataset vs 4.0 for full ComBo
>
> **Detailed model selection results**: We provide details of our selected backbone combinations (models with an "X" are selected) for each dataset with our new approach, and the corresponding test accuracies:
>
> | Dataset | Category | ComBo pruned accuracy | DFN CLIP | DINOv2 | SAM | SigLIP | Models Used |
> |---------|----------|--------------|----------|---------|-----|---------|-------------|
> | CIFAR100 | Natural | 79.4% | - | X | - | - | 1 |
> | Caltech101 | Natural | 94.7% | - | X | - | - | 1 |
> | DTD | Natural | 78.4% | - | X | - | - | 1 |
> | Flowers102 | Natural | 99.7% | - | X | - | - | 1 |
> | Pets | Natural | 93.1% | X | X | - | X | 3 |
> | SVHN | Natural | 85.9% | - | X | - | - | 1 |
> | Sun397 | Natural | 49.3% | - | X | - | - | 1 |
> | Camelyon | Specialized | 86.0% | - | X | - | - | 1 |
> | EuroSAT | Specialized | 96.2% | - | X | - | - | 1 |
> | Resisc45 | Specialized | 91.1% | X | X | - | X | 3 |
> | Retinopathy | Specialized | 73.7% | X | X | - | - | 2 |
> | Clevr-Count | Structured | 92.6% | - | X | - | - | 1 |
> | Clevr-Distance | Structured | 61.9% | X | X | - | X | 3 |
> | DMLab | Structured | 54.3% | X | X | - | - | 2 |
> | KITTI-Distance | Structured | 82.0% | - | X | - | - | 1 |
> | dSprites-Location | Structured | 81.9% | X | - | - | - | 1 |
> | dSprites-Orientation | Structured | 53.3% | - | X | - | - | 1 |
> | SmallNORB-Azimuth | Structured | 36.4% | - | X | - | - | 1 |
> | SmallNORB-Elevation | Structured | 55.6% | - | X | - | - | 1 |
>
> We will also update Table 2 with these new results, with a compact summary shown here:
>
> | Method | Natural | Specialized | Structured | Global Average | Avg. Models/Dataset |
> |--------|---------|-------------|------------|----------------|---------------------|
> | ComBo (DFN CLIP) | 79.3% | 85.8% | 61.6% | 75.6% | 1 |
> | ComBo (DINOv2) | 82.6% | 85.9% | 65.0% | 77.9% | 1 |
> | ComBo (SAM) | 55.2% | 79.4% | 56.9% | 63.8% | 1 |
> | ComBo (SigLIP) | 80.5% | 85.4% | 62.6% | 76.2% | 1 |
> | RADIOv2.5 | 83.8% | 86.6% | 62.1% | 77.5% | 1 |
> | ComBo (all 4 backbones) | 83.3% | 86.5% | 64.8% | 78.2% | 4 |
> | **ComBo (pruned backbones)** | **82.9%** | **86.7%** | **64.8%** | **78.1%** | **1.42** |
>
> We observe that SAM is never selected with our approach, consistent with Table 10, where excluding it improves performance. Thus, our proposed model selection technique provides a solution to the reviewer's Q2.
>
> ### 4. (Q3) Scalability to training examples and classes
>
> **ComBo demonstrates consistent performance gains with increasing training data and effective scaling on the Fungi dataset.**
>
> We agree that validating ComBo's performance in different settings is valuable.
>
> We included results using VTAB full setting subsets for datasets with large training samples in Figure 6 of our appendix. This demonstrates ComBo's performance with increasing training samples (1k, 5k, 10k, 20k), where ComBo consistently benefits from additional data.
>
> To evaluate ComBo's performance on larger class counts, we produced new results on the Fungi dataset [a]. This dataset contains 1,394 species of mushrooms to classify.
>
> [a] Brigit Schroeder and Yin Cui. FGVCx fungi classification challenge 2018. github.com/ visipedia/fgvcx_fungi_comp, 2018.
>
> To observe how performance evolves with class count, we produced subsamples with different numbers of randomly sampled classes and adapted a ViT-B/16 model trained on ImageNet-21K. We present results averaged over three seeds (each leading to different class subsets) compared against baselines:
>
> | Method         | 5 classes | 10 classes | 20 classes | 50 classes | 100 classes | 250 classes | 500 classes | 750 classes | 1000 classes | 1394 classes |
> |----------------|-----------|------------|------------|------------|-------------|-------------|-------------|-------------|--------------|--------------|
> | ComBo          | 77.8%     | 75.6%      | 75.6%      | 62.7%      | 60.8%       | 51.9%       | 44.8%       | 40.7%       | 39.4%        | 36.5%        |
> | Adapter+       | 93.3%     | 88.9%      | 81.1%      | 68.9%      | 66.9%       | 60.3%       | 52.8%       | 47.2%       | 44.3%        | 40.5%        |
> | Linear probing | 84.4%     | 77.8%      | 74.4%      | 63.1%      | 56.0%       | 46.5%       | 37.7%       | 32.2%       | 30.1%        | 26.6%        |
>
> ComBo's behaviour with more classes is comparable to our baselines, with better performance with fewer classes but still decent performance with very high class counts.
>
> ### 5. (Typos) Linear -> affine projection
> We thank the reviewer; this is correct and we will update it in the paper.
>
> ## Summary of updates we will make to the final paper
> Following your feedback, we will include:
> - **Extended architectural generalisation discussion** covering CNN integration possibilities and various tokenisation approaches
> - **Novel automatic model selection methodology and results** (developed in response to multiple reviewers' concerns) showing how our new pruning approach reduces inference costs whilst maintaining performance
> - **Enhanced scalability analysis** including class count scaling experiments
> - **Terminology corrections** (linear → affine projection)

---

> > ### Author Response · Authors · 2025-08-05
> >
> > We thank you once again for your valuable feedback.
> >
> > We have done our best to address your concerns in our rebuttal, and believe that the resulting updates have improved our paper.
> > If you have any remaining questions or comments, we would appreciate it if you could let us know.

---

### Official Review · Reviewer_ARhL · 2025-06-21

**Clarity:** 3
**Significance:** 2
**Originality:** 2
**Rating:** 4
**Confidence:** 5

**Summary:**

The authors present a method to combine the features of multiple pretrained backbones to improve classification accuracy on the VTAB dataset. In particular, the method is a probing based adapter, called ComBo, that utilizes a transformer to compress image features generated from patches at multiple layers from multiple pretrained backbones to a lower dimensional space. ComBo surpasses the performance of competitive probing methods.

**Questions:**

- Have you considered using something other than a linear classifier to predict class probabilities? It has been shown that Linear Discriminant Analysis (LDA) outperforms a linear classifier on VTAB-1K [1]:

[1] Shysheya, A., Bronskill, J., Patacchiola, M., Nowozin, S., & Turner, R. E. (2022). Fit: Parameter efficient few-shot transfer learning for personalized and federated image classification. ICLR 2023.

**Ethical Concerns:**

["NO or VERY MINOR ethics concerns only"]

**Final Justification:**

I will keep my score at 4. The new work on pruning backbones is a nice improvement, but may still be impractical versus fine-tuning with PEFT due to the increase in inference cost.

**Limitations:**

Yes.

**Quality:**

3

**Strengths And Weaknesses:**

**Strengths**
- The paper is well written and easy to read.
- The experiments are thorough and demonstrate the merits of the approach.
- The idea of using a transformer to amalgamate the features from multiple pretrained backbones both spatially and layer-wise is novel.

**Weaknesses**
- The biggest weakness in the approach is the increase in inference time resources (memory, computation) that is proportional to the number of pretrained models $K$. The relatively small accuracy gains (78.2 with ComBo using 4 models from Table 2 versus 77.6 using Adapter+ with a single model) would be difficult to justify in practice given the 4 times increase in inference cost.
- As a new probing method when utilizing just one pretrained backbone, ComBo falls short of PEFT methods (see Table 1).
- The paper would be improved by expanding the evaluation to include few-shot datasets such as Meta-Dataset (Triantafillou, E., Zhu, T., Dumoulin, V., Lamblin, P., Evci, U., Xu, K., ... & Larochelle, H. (2019). Meta-dataset: A dataset of datasets for learning to learn from few examples. arXiv preprint arXiv:1903.03096.)
- A couple of early methods that present various approaches for combining the features of multiple backbones were missed in the references and related work:

    - Chowdhury, A., Jiang, M., Chaudhuri, S., & Jermaine, C. (2021). Few-shot image classification: Just use a library of pre-trained feature extractors and a simple classifier. In Proceedings of the IEEE/CVF International Conference on Computer Vision (pp. 9445-9454).

    - Liu, L., Hamilton, W., Long, G., Jiang, J., & Larochelle, H. (2020). A universal representation transformer layer for few-shot image classification. arXiv preprint arXiv:2006.11702.

---

> ### Author Rebuttal · Authors · 2025-07-31
>
> We thank the reviewer for their positive evaluation and thorough assessment. We appreciate your recognition that the paper is well-written, experiments are thorough, and the transformer-based approach for amalgamating features is novel.
>
> **Overview**: We address your concerns about inference cost vs. accuracy trade-offs, performance relative to PEFT methods, and provide additional experimental analysis on alternative classifiers, while acknowledging suggestions for future work.
>
> ## Replies to your concerns and questions
> ### 1. Inference cost being proportional to the number of models
>
> **Our new automatic backbone pruning reduces the average number of backbones from 4 to 1.42 whilst maintaining performance (78.1% vs 78.2%).**
>
> In direct response to concerns about inference costs of using multiple backbones raised by multiple reviewers, we have developed and evaluated a novel automatic backbone pruning method that directly addresses the $K$× inference cost issue.
>
> Our method reduces average backbones per dataset from 4.0 to 1.42 while maintaining competitive performance (78.1% vs 78.2% for full ComBo).
>
> ### Methodology: Two-stage training approach
> Our method is inspired by Head2Toe [11], but instead of selecting features to retain, it selects entire backbones.
>
> **Stage 1 - Learning backbone importance**: We first train ComBo with a novel regularisation term applied to projection weights Λ. For each backbone model, we compute the L2 norm of all its projection weights across all layers to obtain an importance score. We sum importance scores from all backbones and add the result as a regularisation term to minimise in our loss.
>
> **Stage 2 - Pruning and retraining**: After the first stage of training, we measure the resulting importance scores for each backbone. We prune all backbones whose norms fall below 90% of the highest-scoring backbone, then retrain ComBo from scratch without regularisation using only the selected backbones.
>
> ### Experimental results
> With our novel pruning mechanism:
> - Performance is maintained: ComBo with pruning achieves 78.1% vs 78.2% for full ComBo (only 0.1% drop)
> - Efficiency is improved: On average over the 19 datasets, we select 1.42 backbones per dataset vs 4.0 for full ComBo
>
> We provide details of our selected backbone combinations (models with an "X" are selected) for each dataset with our new approach, and the corresponding test accuracies:
>
> | Dataset | Category | ComBo pruned accuracy | DFN CLIP | DINOv2 | SAM | SigLIP | Models Used |
> |---------|----------|--------------|----------|---------|-----|---------|-------------|
> | CIFAR100 | Natural | 79.4% | - | X | - | - | 1 |
> | Caltech101 | Natural | 94.7% | - | X | - | - | 1 |
> | DTD | Natural | 78.4% | - | X | - | - | 1 |
> | Flowers102 | Natural | 99.7% | - | X | - | - | 1 |
> | Pets | Natural | 93.1% | X | X | - | X | 3 |
> | SVHN | Natural | 85.9% | - | X | - | - | 1 |
> | Sun397 | Natural | 49.3% | - | X | - | - | 1 |
> | Camelyon | Specialized | 86.0% | - | X | - | - | 1 |
> | EuroSAT | Specialized | 96.2% | - | X | - | - | 1 |
> | Resisc45 | Specialized | 91.1% | X | X | - | X | 3 |
> | Retinopathy | Specialized | 73.7% | X | X | - | - | 2 |
> | Clevr-Count | Structured | 92.6% | - | X | - | - | 1 |
> | Clevr-Distance | Structured | 61.9% | X | X | - | X | 3 |
> | DMLab | Structured | 54.3% | X | X | - | - | 2 |
> | KITTI-Distance | Structured | 82.0% | - | X | - | - | 1 |
> | dSprites-Location | Structured | 81.9% | X | - | - | - | 1 |
> | dSprites-Orientation | Structured | 53.3% | - | X | - | - | 1 |
> | SmallNORB-Azimuth | Structured | 36.4% | - | X | - | - | 1 |
> | SmallNORB-Elevation | Structured | 55.6% | - | X | - | - | 1 |
>
> We will also update Table 2 with these new results, with a compact summary shown here:
>
> | Method | Natural | Specialized | Structured | Global Average | Avg. Models/Dataset |
> |--------|---------|-------------|------------|----------------|---------------------|
> | ComBo (DFN CLIP) | 79.3% | 85.8% | 61.6% | 75.6% | 1 |
> | ComBo (DINOv2) | 82.6% | 85.9% | 65.0% | 77.9% | 1 |
> | ComBo (SAM) | 55.2% | 79.4% | 56.9% | 63.8% | 1 |
> | ComBo (SigLIP) | 80.5% | 85.4% | 62.6% | 76.2% | 1 |
> | RADIOv2.5 | 83.8% | 86.6% | 62.1% | 77.5% | 1 |
> | ComBo (all 4 backbones) | 83.3% | 86.5% | 64.8% | 78.2% | 4 |
> | ComBo (pruned backbones) | 82.9% | 86.7% | 64.8% | 78.1% | 1.42 |
>
>
> ### 2. Performance vs. PEFT methods
>
> **ComBo enables multi-model combination and requires 4× less GPU memory than PEFT methods during training.**
>
> We agree that parameter-efficient fine-tuning (PEFT) techniques are strong baselines that ComBo might not usually outperform in the context of adapting a single backbone. However, we designed our approach to efficiently combine features from multiple models, which PEFT methods currently do not handle naturally.
>
> Moreover, not requiring backpropagation through the backbone offers other advantages, even with a single backbone. We have measured GPU memory requirements during training (which we will include in the paper) and found that ComBo requires 4× less memory than Adapter+. This property can make ComBo a valuable choice for practitioners with limited computational resources.
>
> As a practical example, in our setting with an NVIDIA RTX 3090 Ti (24GB) GPU, we are able to probe the Giant version of DINOv2 (ViT-g/14) with ComBo, but are unable to adapt it with Adapter+ due to memory limitations without needing additional tricks (e.g., accumulating gradients).
>
> ### 3. Future work and missing references
> **We will include missing references and discuss few-shot learning capabilities studies as future work.**
>
> - Few-shot setting evaluation: We thank the reviewer for this suggestion and agree that validating ComBo's few-shot learning capabilities on a benchmark such as Meta-Dataset would be an interesting next step. Unfortunately, it was outside the scope of our current project and submission, but we will consider it in future work.
> - Missing references: We thank the reviewer for pointing out these very relevant methods we had missed in our related work. We will include them in the revised manuscript.
>
>
> ### 4. Alternative classification head
>
> **Linear head consistently outperforms Naive Bayes classifier across all task categories in our setting.**
>
> The previous observation from the FiT paper on an improved classification head on VTAB-1k is interesting, and we have now performed an additional ablation study to see its effect in our setting. We replaced the linear head in ComBo with a Gaussian Naive Bayes classifier following the approach used in the FiT-LDA method.
>
> We obtained the following results (same settings as Table 5: averaged over three seeds, on the validation set of VTAB-1k), which we will include in our paper:
> | Classifier Type | Natural Tasks | Specialized Tasks | Structured Tasks | Average |
> |-----------------|---------------|-------------------|------------------|---------|
> | ComBo with linear head | 79.3% | 85.6% | 56.5% | 73.8% |
> | ComBo with FiT-LDA head | 77.3% | 69.8% | 54.8% | 67.3% |
>
> We find that the linear head consistently outperforms the Naive Bayes classifier across all task categories. This differs from FiT's reported improvements, potentially due to our use of non-episodic training, which is typical in transfer learning, whereas the authors of FiT noted that episodic fine-tuning was crucial to obtain optimal Naive Bayes performance.
>
> ## Summary of updates we will make to the final paper
>
> Following your feedback, we will include:
> - Novel automatic model selection methodology and results (developed in response to multiple reviewers' concerns) showing how our new pruning approach reduces inference costs whilst maintaining performance
> - GPU memory requirement details for training ComBo and PEFT methods and discussion highlighting accessibility advantages for practitioners with limited computational resources during training
> - Comparison to the Gaussian Naive Bayes classifier from FiT-LDA in our ablation study
> - Additional citations with proper contextualisation of related feature combination approaches
> - Future work discussion of studying ComBo's potential in few-shot learning settings

---

> > ### Comment · Reviewer_ARhL · 2025-08-02
> >
> > Thanks for the response.
> >
> > Regarding: “Our new automatic backbone pruning reduces the average number of backbones from 4 to 1.42 whilst maintaining performance (78.1% vs 78.2%).”
> >
> > It is good that you can potentially reduce the number of backbones via pruning. However, stating the number to be 1.42 is a bit misleading since you are pruning entire backbones, not fractions of backbones. I understand that this is an average over the VTAB datasets, however, if I understand correctly, the pruning would often reduce the number of backbones to one or two. If it is one, extra inference cost is not an issue. If it is two or more, the inference, memory, and compute cost will be on the order of 2x, which may make PEFT a better alternative.

---

> > > ### Author Response · Authors · 2025-08-04
> > >
> > > Thank you for your consideration of our rebuttal and acknowledgment of the value of our new pruning approach.
> > >
> > > We understand your concern and see how presenting the average over all datasets might be misleading. We will ensure to **explicitly clarify this point in the paper**, specifying that depending on the dataset, our relative threshold-based pruning mechanism selects between 1 and 3 backbones to keep.
> > >
> > > Moreover, motivated by your comment, we also plan to **include a "budget-based" selection criterion** that we believe would be more practically representative. In practice, users might be limited to a maximum number of N backbones due to their computational budget at deployment time. Therefore, instead of using a relative threshold to select models to keep (which leads to a variable number of models), we can select the top N backbones based on our model pruning mechanism.
> > >
> > > This approach would lead to the following results on the test set, where we can clearly see the performance for each specified number of backbones selected:
> > >
> > > | Method | Natural | Specialized | Structured | Global Average | Number of backbones used |
> > > |--------|---------|-------------|------------|----------------|---------------------|
> > > | ComBo - DFN CLIP | 79.3% | 85.8% | 61.6% | 75.6% | 1 |
> > > | ComBo - DINOv2 | 82.6% | 85.9% | 65.0% | 77.9% | 1 |
> > > | ComBo - SAM | 55.2% | 79.4% | 56.9% | 63.8% | 1 |
> > > | ComBo - SigLIP | 80.5% | 85.4% | 62.6% | 76.2% | 1 |
> > > | RADIOv2.5 | 83.8% | 86.6% | 62.1% | 77.5% | 1 |
> > > | ComBo - all 4 backbones | 83.3% | 86.5% | 64.8% | 78.2% | 4 |
> > > | ComBo - Pruning to keep top 1 backbone | 82.8% | 86.0% | 64.3% | 77.7% | 1 |
> > > | ComBo - Pruning to keep top 2 backbones | 84.0% | 86.3% | 65.3% | 78.6% | 2 |
> > > | ComBo - Pruning to keep top 3 backbones | 83.7% | 86.5% | 64.9% | 78.4% | 3 |
> > >
> > > These results also highlight the potential of our pruning method to improve performance by discarding backbones that do not contribute to performance gains.
> > >
> > > We hope that this presentation of results based on a "budget-based" selection using our pruning approach will prevent any confusion about inference costs and be more practically useful, while clearly allowing users to consider trade-offs against PEFT methods.

---

### Official Review · Reviewer_UTfc · 2025-07-03

**Clarity:** 3
**Significance:** 2
**Originality:** 2
**Rating:** 4
**Confidence:** 4

**Summary:**

Authors propose ComBo, a method to combine features from multiple foundation models. Specifically a two-stage approach is adopted: features from multiple layers and models are concatenated and projected, which is then passed into a lightweight transformer for further processing. Experiment results show that ComBo is better than other baselines like Head2Toe and SMP.

**Questions:**

see weaknesses

**Ethical Concerns:**

["NO or VERY MINOR ethics concerns only"]

**Final Justification:**

I notice that in the new result provided in rebuttal, Dinov2 is selected 18 out of 19 cases. And for all but 1 single-backbone cases, DINOv2 is the choice. It seems that the conclusion from the current set of experiments boils down to "just use DINOv2", as also shown by the high accuracy by Combo+DINOv2 (77.9%). This result seems to defeat the paper's story on combining multiple backbones, if one single backbone can achieve 77.9%, there is less reason to add in other backbones for ~0.4% accuracy improvement. Therefore, the current experiment setting where one vision backbone is dominating may not be enough to show ComBo's motivation on leveraging strength of multiple backbones.

=======edit========

Author's additional results show the informativeness of ComBo as a criteria to select and prune multiple pretrained vision backbones. I raise my score to 4.

**Quality:**

2

**Strengths And Weaknesses:**

Strength:
1. Overall the paper is easy to follow.
2. The proposed method ComBo does not require extensive hyperparameter tuning.

Weakness:
1. The method proposed is not practical. When combining features from multiple models, it requires forwarding multiple models and obtaining their features. In Table 6, if one puts the distillation baseline in it, it should be far more efficient than ComBo since it does not require multiple models’ inference.
2. When compared with LoRA, the proposed method has worse performance while being less efficient during inference time. Admittedly, ComBo does not require back propagation through the foundation model. However I doubt if the real world has cases where trading inference’s speed for training efficiency is needed. I encourage authors to justify the real-world use case here.
3. In spite of the limited practicality, the ideas raised in future work sections are potentially interesting and it would add in this paper’s value if some of the analysis could be done: e.g. analyzing the parameters to help with model selection/pruning. This may bring insight on understanding different FMs, adding to the value of this work.

---

> ### Author Rebuttal · Authors · 2025-07-31
>
> We thank the reviewer for their feedback and for acknowledging that the paper is easy to follow and that the method does not require extensive hyperparameter tuning. We appreciate your constructive concerns about practical applications, which have motivated us to develop novel solutions that directly address inference efficiency.
>
> **Overview**: We address your main concerns about practicality (Weaknesses 1-2) by clarifying the training vs. inference efficiency trade-offs, providing new model pruning results (Weakness 3) that reduce inference costs, and demonstrating ComBo's advantages in memory-constrained scenarios.
>
> ## Replies to your concerns and questions
> ### 1. (Weakness 1) Inference cost against a distillation baseline
>
> **ComBo addresses cases where distillation is impractical due to resource constraints, whilst offering automatic model selection to reduce inference costs.**
>
> While we agree that a distilled model would be more efficient at inference for combining multiple models than ComBo, which still requires forward passes from each backbone, we believe there is a strong need for an approach such as ComBo.
>
> The resources required to distil foundation models are extremely significant, with SAM-CLIP [4] requiring 64 A100 GPUs, and RADIOv2.5 [5] also using 64 GPUs to train over 614M examples. These resources are not accessible to many practitioners, who would need to rely on distilled models being released. If a new foundation model relevant to their tasks becomes available, it would be costly or impossible for many practitioners to include it in a distillation process. In contrast, ComBo can be trained in ~10 minutes to probe 4 models on a VTAB-1k task, using a standard consumer GPU.
>
> Moreover, distillation is not a straightforward process, with potential issues such as "mode switches" appearing based on resolution [5].
>
> Finally, our additional pruning approach (addressing your Weakness #3, detailed below) allows us to automatically select a subset of backbones for a given task, thereby reducing the inference cost significantly.
>
> ### 2. (Weakness 2) Comparison against LoRA
>
> **ComBo enables multi-model combination and requires 4× less GPU memory than PEFT methods during training.**
>
> We agree that LoRA and other parameter-efficient fine-tuning (PEFT) techniques are often very practical, offering strong performance with minor inference cost overheads.
>
> One key advantage of our probing approach is that it does not require backpropagating through the backbone. If this is manageable, we agree that PEFT methods might be a better choice. However, our objective is to access features from multiple, potentially large, backbones. LoRA and other PEFT techniques are not designed for this setting, and the need to backpropagate through the backbones strongly limits which backbones can be combined. In contrast, ComBo is specifically designed to make best use of multiple models efficiently.
>
> Even in the single backbone case, the GPU memory requirements can make ComBo a valuable choice for practitioners with limited computational resources. We have measured GPU memory requirements during training (which we will include in the paper) and found that ComBo requires 4× less memory than Adapter+.
>
> As a practical example, in our setting with an NVIDIA RTX 3090 Ti (24GB) GPU, we are able to probe the Giant version of DINOv2 (ViT-g/14) with ComBo, but are unable to adapt it with Adapter+ due to memory limitations without needing additional tricks (e.g., accumulating gradients).
>
> Finally, ComBo's inference cost is slightly higher than some PEFT methods, but we find the difference is very minor in practice (<3% additional FLOPs relative to the original backbone).
>
>
> ### 3. (Weakness 3) Novel Model Pruning Method (Developed in Response to Multiple Reviewers)
>
> **Our new automatic backbone selection reduces the average number of backbones from 4 to 1.42 whilst maintaining performance (78.1% vs 78.2%).**
>
> In direct response to concerns about inference efficiency raised by you and other reviewers, we have developed and evaluated a novel automatic model pruning method that significantly reduces inference costs while maintaining performance.
>
> Our two-stage method is inspired by Head2Toe [11], but instead of selecting features to retain, it selects entire backbones.
>
> **Stage 1 - Learning Backbone Importance**: We first train ComBo with a novel regularisation term applied to projection weights Λ. For each backbone model, we compute the L2 norm of all its projection weights across all layers to obtain an importance score. We sum importance scores from all backbones and add the result as a regularisation term to minimise in our loss.
>
> **Stage 2 - Pruning and Retraining**: After the first stage of training, we measure the resulting importance scores for each backbone. We prune all backbones whose norms fall below 90% of the highest-scoring backbone, then retrain ComBo from scratch without regularisation using only the selected backbones.
>
> ### Experimental Results
> With our new pruning mechanism:
> - Performance is maintained: ComBo with pruning achieves 78.1% vs 78.2% for full ComBo (only 0.1% drop)
> - Efficiency is improved: On average over the 19 datasets, we select 1.42 backbones per dataset vs 4.0 for full ComBo
>
> We provide details of our selected backbone combinations (models with an "X" are selected) for each dataset with our new approach, and the corresponding test accuracies:
>
> | Dataset | Category | ComBo pruned accuracy | DFN CLIP | DINOv2 | SAM | SigLIP | Models Used |
> |---------|----------|--------------|----------|---------|-----|---------|-------------|
> | CIFAR100 | Natural | 79.4% | - | X | - | - | 1 |
> | Caltech101 | Natural | 94.7% | - | X | - | - | 1 |
> | DTD | Natural | 78.4% | - | X | - | - | 1 |
> | Flowers102 | Natural | 99.7% | - | X | - | - | 1 |
> | Pets | Natural | 93.1% | X | X | - | X | 3 |
> | SVHN | Natural | 85.9% | - | X | - | - | 1 |
> | Sun397 | Natural | 49.3% | - | X | - | - | 1 |
> | Camelyon | Specialized | 86.0% | - | X | - | - | 1 |
> | EuroSAT | Specialized | 96.2% | - | X | - | - | 1 |
> | Resisc45 | Specialized | 91.1% | X | X | - | X | 3 |
> | Retinopathy | Specialized | 73.7% | X | X | - | - | 2 |
> | Clevr-Count | Structured | 92.6% | - | X | - | - | 1 |
> | Clevr-Distance | Structured | 61.9% | X | X | - | X | 3 |
> | DMLab | Structured | 54.3% | X | X | - | - | 2 |
> | KITTI-Distance | Structured | 82.0% | - | X | - | - | 1 |
> | dSprites-Location | Structured | 81.9% | X | - | - | - | 1 |
> | dSprites-Orientation | Structured | 53.3% | - | X | - | - | 1 |
> | SmallNORB-Azimuth | Structured | 36.4% | - | X | - | - | 1 |
> | SmallNORB-Elevation | Structured | 55.6% | - | X | - | - | 1 |
>
> We will also update Table 2 with these new results, with a compact summary shown here:
>
> | Method | Natural | Specialized | Structured | Global Average | Avg. Models/Dataset |
> |--------|---------|-------------|------------|----------------|---------------------|
> | ComBo (DFN CLIP) | 79.3% | 85.8% | 61.6% | 75.6% | 1 |
> | ComBo (DINOv2) | 82.6% | 85.9% | 65.0% | 77.9% | 1 |
> | ComBo (SAM) | 55.2% | 79.4% | 56.9% | 63.8% | 1 |
> | ComBo (SigLIP) | 80.5% | 85.4% | 62.6% | 76.2% | 1 |
> | RADIOv2.5 | 83.8% | 86.6% | 62.1% | 77.5% | 1 |
> | ComBo (all 4 backbones) | 83.3% | 86.5% | 64.8% | 78.2% | 4 |
> | ComBo (pruned backbones) | 82.9% | 86.7% | 64.8% | 78.1% | 1.42 |
>
>
> ## Summary of updates we will make to the final paper
> Following your feedback, we will include:
> - Expanded computational details detailing training costs of distillation and PEFT techniques compared to ComBo, including detailed memory usage comparisons
> - Enhanced practical use case discussion highlighting scenarios where the computational trade-offs offered by ComBo are valuable
> - Novel automatic model selection methodology and results (developed in response to multiple reviewers' concerns) showing how our new pruning approach reduces inference costs whilst maintaining performance

---

> > ### Comment · Reviewer_UTfc · 2025-08-03
> > **Limited improvement over single backbone: ComBo + Single Backbone achieves 77.9%, whereas multiple backbones are just 78.2%**
> >
> > Thank authors for the rebuttal and additional experiments. Here are my questions:
> >
> > > Finally, ComBo's inference cost is slightly higher than some PEFT methods, but we find the difference is very minor in practice (<3% additional FLOPs relative to the original backbone).
> >
> > I believe 3% is the additional overhead that is brought by the added transformer architecture. However, my main concern is that when two or more backbones are employed, the inference time is ~2x (assuming these backbones are approximately of the same FLOPs) the original since if someone is going to use it.
> >
> > > Our new automatic backbone selection reduces the average number of backbones from 4 to 1.42 whilst maintaining performance (78.1% vs 78.2%).
> >
> > Thanks authors for this additional experiment. But I notice that Dinov2 is selected 18 out of 19 cases. And for all but 1 single-backbone cases, DINOv2 is the choice. It seems that the conclusion from the current set of experiments boils down to "just use DINOv2", as also shown by the high accuracy by Combo+DINOv2 (77.9%). This result seems to defeat the paper's story on combining multiple backbones, if one single backbone can achieve 77.9%, there is less reason to add in other backbones for ~0.4% accuracy improvement. Therefore, the current experiment setting where one vision backbone is dominating may not be enough to show ComBo's motivation on leveraging strength of multiple backbones.
> >
> > On the other hand, I see authors exploration as a potential lightweight single model selection approach. (However, this is different from the existing story, and may need to compare with different baselines).
> >
> > In light of these, I will currently keep my rating.

---

> > > ### Author Response · Authors · 2025-08-04
> > >
> > > Thank you for your thoughtful consideration of our rebuttal and the valid points you raise.
> > >
> > > ### About the inference cost against PEFT
> > > Thank you for the clarification; we understand your concern and agree with your assessment. We acknowledge that when multiple backbones are used, inference cost will be higher than PEFT methods with a single backbone. However, we still believe that, depending on a user's task and computational budget, ComBo's unique ability to combine multiple diverse backbones can unlock improved performance while remaining computationally acceptable.
> > >
> > > ### About the dominance of a single model (DINOv2)
> > > Thank you for this observation. It is true that DINOv2 appears to be the strongest individual model on VTAB overall, and our pruning mechanism's selection presented in our rebuttal reflects this dominance.
> > >
> > > However, we believe this stems from our conservative choice of a 90% relative threshold for model selection. We made this choice to demonstrate the potential for significantly reducing the number of backbones, but we find that performance is often maximised when combining multiple models.
> > >
> > > To highlight this, we have performed an exhaustive evaluation of all combinations of models on the VTAB-1k validation set and present the best-performing combinations for each dataset:
> > >
> > > | Dataset | Category | DFN CLIP | DINOv2 | SAM | SigLIP | Models Used |
> > > |---------|----------|----------|---------|-----|---------|-------------|
> > > | CIFAR100 | Natural | X | X | - | - | 2 |
> > > | Caltech101 | Natural | - | X | X | - | 2 |
> > > | DTD | Natural | X | X | - | X | 3 |
> > > | Flowers102 | Natural | - | - | - | X | 1 |
> > > | Pets | Natural | - | - | - | X | 1 |
> > > | SVHN | Natural | X | X | X | X | 4 |
> > > | Sun397 | Natural | - | - | - | X | 1 |
> > > | Camelyon | Specialized | - | X | X | - | 2 |
> > > | EuroSAT | Specialized | X | X | - | X | 3 |
> > > | Resisc45 | Specialized | - | X | - | X | 2 |
> > > | Retinopathy | Specialized | - | X | - | X | 2 |
> > > | Clevr-Count | Structured | - | X | - | X | 2 |
> > > | Clevr-Distance | Structured | X | X | X | X | 4 |
> > > | DMLab | Structured | X | X | - | X | 3 |
> > > | KITTI-Distance | Structured | - | X | - | - | 1 |
> > > | dSprites-Location | Structured | X | X | X | X | 4 |
> > > | dSprites-Orientation | Structured | - | X | - | - | 1 |
> > > | SmallNORB-Azimuth | Structured | X | X | - | - | 2 |
> > > | SmallNORB-Elevation | Structured | - | X | - | - | 1 |
> > >
> > > Below, we present the resulting validation-set performance on the VTAB-1k benchmark. As baselines, we also include the average performance when randomly selecting a combination of N models. Please note that these results are on the VTAB-1k validation set and cannot be directly compared to the numbers from the original table in our rebuttal, which are on the test set.
> > >
> > > | Backbone selection method | Natural | Specialized | Structured | Global Average |
> > > |--------|---------|-------------|------------|----------------|
> > > | ComBo - DINOv2 backbone only | 80.7% | 86.2% | 61.1% | 76.0% |
> > > | ComBo - All four backbones | 81.8% | 86.4% | 60.3% | 76.2% |
> > > | ComBo - Best-performing combination from exhaustive search | 83.6% | 88.1% | 64.0% | 78.6% |
> > > | ComBo - Pruning with the 90% relative threshold | 80.9% | 87.2% | 61.9% | 76.7% |
> > > | ComBo - Randomly selecting 1 backbone | 73.5% | 84.8% | 58.2% | 72.2% |
> > > | ComBo - Randomly selecting 2 backbones | 80.2% | 86.2% | 59.7% | 75.4% |
> > > | ComBo - Randomly selecting 3 backbones | 81.4% | 86.4% | 60.6% | 76.1% |
> > >
> > > These results validate the potential benefits of combining multiple models and indicate that the optimal combination is dataset-specific. The results demonstrate that ComBo's ability to leverage optimal backbone combinations has the potential to strongly outperform DINOv2 alone (78.6% vs. 76.0% with ideal combinations).
> > >
> > > While our selection mechanism does not yet achieve the same performance as exhaustive search, it still makes reasonably informed selections that outperform random selections.
> > >
> > > We will update the final paper to include these comprehensive results and discuss:
> > > - The potential for improved performance through ideal model combinations, which are dataset-specific and generally involve multiple models rather than relying on a single best model
> > > - The effectiveness of our pruning mechanism, while acknowledging room for improvement in future work to bridge the gap between the ideal combination selection discovered through exhaustive search and our current approach

---

> > > > ### Comment · Reviewer_UTfc · 2025-08-05
> > > >
> > > > Thank the authors for the additional experiments. I do believe this experiment justifies the motivation for combining multiple backbones by showing the results from the exhaustive search. That said, there is a significant gap between the best-performing combination and pruning with a 90% relative threshold (76.7% vs. 78.6%).
> > > >
> > > > From the pruning criteria, there should be a ranking of the four backbones for each dataset (please confirm this). Can the authors show whether this ranking aligns with the chosen backbones in the exhaustive search? Based on this ranking, for each dataset, assuming one can select the top k backbones (where k ranges from 1 to 4), and pick the k resulting in the best performance for each dataset—can the authors perhaps show this accuracy instead? This experiment would demonstrate whether the resulting ComBo weights are indeed informative for backbone selection.
> > > >
> > > > I believe whether the pruning criteria accurately reflects the benefit of combining backbones is important, especially in light of the validation results where using all four backbones (76.2%) is significantly worse than choosing fewer backbones (78.6%).

---

> > > > > ### Author Response · Authors · 2025-08-05
> > > > >
> > > > > Thank you for the insightful suggestion.
> > > > >
> > > > > You are correct that our pruning mechanism provides a ranking of the models, and we agree that the proposed experiment would provide an excellent validation of this ranking.
> > > > >
> > > > > We conducted the suggested experiment and verified the performance obtained when, for each dataset, we rank backbones using our selection mechanism and select the top k backbones such that k results in the best performance. The results are presented in the updated table below:
> > > > > | Backbone selection method | Natural | Specialized | Structured | Global Average |
> > > > > |--------|---------|-------------|------------|----------------|
> > > > > | ComBo - DINOv2 backbone only | 80.7% | 86.2% | 61.1% | 76.0% |
> > > > > | ComBo - All four backbones | 81.8% | 86.4% | 60.3% | 76.2% |
> > > > > | ComBo - Best-performing combination from exhaustive search | 83.6% | 88.1% | 64.0% | 78.6% |
> > > > > | ComBo - Pruning with the 90% relative threshold | 80.9% | 87.2% | 61.9% | 76.7% |
> > > > > | ComBo - Randomly selecting 1 backbone | 73.5% | 84.8% | 58.2% | 72.2% |
> > > > > | ComBo - Randomly selecting 2 backbones | 80.2% | 86.2% | 59.7% | 75.4% |
> > > > > | ComBo - Randomly selecting 3 backbones | 81.4% | 86.4% | 60.6% | 76.1% |
> > > > > | ComBo - Selecting top k backbones based on our mechanism's ranking (where k depends on the dataset) | 83.3% | 87.4% | 63.7% | 78.1% |
> > > > >
> > > > > These results validate that our mechanism's ranking provides informative guidance, achieving a global average of 78.1%, which comes within 0.5 percentage points of the upper bound established through exhaustive (but impractical) combination search. This also substantially outperforms baselines such as using all available backbones, only the single best backbone, or randomly selecting combinations of backbones.
> > > > >
> > > > > Overall, your feedback motivated us to: (1) develop an automatic backbone selection method, (2) better validate both this mechanism and the benefits of combining multiple backbones, and (3) better articulate practical considerations relative to other adaptation and distillation techniques.
> > > > > We believe these improvements have substantially enhanced our paper's completeness and ComBo's practical applicability while highlighting the value in its unique capability to probe features from multiple foundation models.
> > > > >
> > > > > We hope our updates have thoroughly addressed the concerns identified in our original submission.
> > > > > We thank you once again for your valuable comments, and please do not hesitate to let us know if there are any remaining concerns.

---

> > > > > > ### Comment · Reviewer_UTfc · 2025-08-05
> > > > > >
> > > > > > Thanks authors for the results provided. I believe this is concrete evidence of the usefulness of ComBo in selecting and combining models. I encourage authors to further think about how to reach this 78.1% without needing to search "k“。
> > > > > >
> > > > > > I suggest the additional pruning/selection experiment be included as motivation and the main body of the paper, which I think is of more practical value than combining all four backbones.
> > > > > >
> > > > > > In light of this, I will raise my score to 4 (borderline accept).

---

> > > > > > > ### Author Response · Authors · 2025-08-07
> > > > > > >
> > > > > > > Thank you for your reply and for reconsidering your original score.
> > > > > > >
> > > > > > > We appreciate your suggestions, and will further think about improving the method to find the optimal number of backbones.
> > > > > > > We will also update our paper following your suggestion to discuss model selection based on our model combination experiments.

---

### Official Review · Reviewer_YFvs · 2025-07-04

**Clarity:** 3
**Significance:** 3
**Originality:** 3
**Rating:** 4
**Confidence:** 1

**Summary:**

Foundation models learn diverse features based on their training data and objectives. For downstream tasks, different models or layers within these models may provide useful representations. Existing strategies such as fine-tuning and distillation are unable to exploit these complementary strengths or are too expensive. In this paper, the authors present ComBo, a lightweight, scalable adapter based on the probing framework that (i) extracts feature representations from selected layers of multiple models, (ii) normalizes and aligns them spatially via interpolation, (iii) projects them into low-dimensional embeddings using a shared linear projection, and (iv) processes the compressed embeddings with a small transformer. The method operates on frozen models and does away with the expensive backprop into FMs. It also avoids dataset-specific tuning. The approach was tested on 19 vision tasks across natural, specialized, and structured domains. ComBo beats previous probing methods (Head2Toe, SMP), and matches or surpasses full fine-tuning in many structured tasks. It also beats distilled models like RADIOv2.5.

**Questions:**

The paper assumes that spatial features from different foundation models can be aligned via interpolation to a shared spatial resolution. This may work well for ViT-style models, but how generalizable or robust this assumption is, particularly for architectures with fundamentally different tokenization strategies.

**Ethical Concerns:**

["NO or VERY MINOR ethics concerns only"]

**Final Justification:**

Thank you for the detailed response. I believe the revisions suggested will strengthen the final version.

**Limitations:**

yes

**Quality:**

3

**Strengths And Weaknesses:**

Strengths:

- doesn't require backprob through foundation models making the approach computationally efficient
- handles activations from multiple models and layers without task-specific tuning unless previous probing methods
- outperforms existing probing approaches and matches/surpasses full fine-tuning in structured tasks
- uses fixed hyperparameters making it robust and easier to apply
- works with both pretrained or tned models and integrates new models without any retraining
- lower parameter overhead compared to multi-backbone fine-tuning and distillation

Weaknesses:

- The paper assumes that spatial features from different foundation models can be aligned via interpolation to a shared spatial resolution. This may work well for ViT-style models, but how generalizable or robust this assumption is, particularly for architectures with fundamentally different tokenization strategies.
- novelty of the paper is more system integration than in algorithmic design

---

> ### Author Rebuttal · Authors · 2025-07-31
>
> We thank the reviewer for their positive evaluation and constructive feedback. We appreciate your recognition of ComBo's key strengths: computational efficiency, robustness without task-specific tuning, and ability to integrate multiple models without retraining each separately.
>
> **Overview**: We address your concerns about generalisability to different architectures and clarify the novelty and practical contributions of our approach.
>
> ## Replies to your concerns and questions
> ### 1. Generalisability to different architectures and tokenisation strategies
>
> We thank the reviewer for pointing out that this was not discussed. We agree that this aspect and the requirements to support new architectures should be clarified in the paper, and we will include a discussion about it.
>
> **ViT-based approaches are standard**
>
> Whilst we agree that our experiments consider limited architectures, we believe our findings remain generally useful.
> Our current evaluation focuses on ViT-based architectures as they represent the dominant paradigm in modern foundation models. As such, we expect the application of ComBo to remain straightforward for most recent, and potentially future, foundation models.
>
> **The general methodology of ComBo could be adapted to other architectures**
>
> Moreover, the core principle of ComBo can be extended beyond this specific architecture family: The core methodology remains (1) extract features from multiple layers of multiple models, (2) align them spatially in an architecture-dependent manner, (3) compress them via a learned projection, (4) process the compressed features with a transformer. The spatial alignment step can be customised for different architectures whilst maintaining the overall framework.
>
> **Tokenisation discrepancies and Convolutional Neural Networks (CNNs) could be managed through interpolation**
>
> Whilst tokenisation discrepancies might appear within ViT-based models, we expect some spatial alignment to be possible. In fact, this is the case in our experiments, where DINOv2 uses 14×14 patch sizes compared to 16×16 for the other models we consider. This required us to combine the models by spatially interpolating their resolution, which produced a simple and successful combination strategy.
>
> We believe CNN feature maps can likely be dealt with similarly, by the same interpolation approach on different layers, treating spatial locations as tokens.
>
> ### 2. Addressing novelty
>
> **ComBo demonstrates novel insights about multi-model feature integration and provides practical capabilities not previously shown.**
>
> We agree that ComBo represents system integration rather than purely algorithmic design. However, we believe this system provides sufficient novelty through the insights it generates and its practical contributions:
>
> ### Novel insights from our work
> - Transformers can effectively process multi-model features: We demonstrate that transformers can successfully integrate features from multiple layers across different foundation models, even in a low-data setting such as VTAB-1k.
> - Different features from different foundation models can be relevant, depending on the task: Our results highlight the importance of selecting the correct foundation model and layer to extract features for a given task.
>
> ### Practical novelty
> Whilst individual components of ComBo exist, the design of ComBo enables capabilities not previously demonstrated:
> - Probing multiple large foundation models together without backpropagation through any of them
> - Universal hyperparameters that work across diverse vision tasks
> - Memory-efficient approach that makes large model combinations accessible to practitioners with limited resources
>
> ### Value of simple systems
> We argue that demonstrating how simple integration, albeit the result of significant design exploration, can match complex alternatives represents valuable novelty for the field. Our work provides practical insights about what components are necessary for effective multi-model combination, which we believe has clear value for the community.
>
> ## Summary of updates we will make to the final paper
> Following your feedback, we will include:
> - Extended architectural generalisation discussion covering the application of ComBo to other backbone architectures and various tokenisation approaches
> - Clarifications on the novelty of our work, discussing its design choices and strengths in its simple architecture

---

> > ### Comment · Reviewer_YFvs · 2025-08-01
> > **Re: Rebuttal by Authors**
> >
> > Thank you for the detailed response. I believe the revisions suggested will strengthen the final version.

---

### Decision · Program_Chairs · 2025-09-17

**Decision:**

Accept (poster)

**Comment:**

The paper introduces a probing method for combining features from multiple foundation models, aiming to leverage complementary strengths across modalities and architectures. Reviewers agree that the paper makes a focused contribution, backed by solid experimental results. Key weaknesses are the scope of experiments (e.g., limited to certain datasets and modalities) and the risk of not generalizing to more complex settings. The rebuttal provided additional evidence and new results, resolving some concerns. Overall, the strengths overweigh weaknesses, and I recommend to accept this paper.